# Velocity variations and hydrological drainage at Baltoro Glacier, Pakistan

Anna Wendleder[1], Jasmin Bramboeck[2], Jamie Izzard[3], Thilo Erbertseder[1], Pablo d'Angelo[4], Andreas Schmitt[2], Duncan J. Quincey[3], Christoph Mayer[5], and Matthias H. Braun[6]

[1]German Remote Sensing Data Center, German Aerospace Center, Oberpfaffenhofen, Germany
[2]Institute for Applications of Machine Learning and Intelligent Systems, Munich University of Applied Sciences, Munich, Germany
[3]School of Geography, University of Leeds, Leeds, UK
[4]Remote Sensing Technology Institute, German Aerospace Center, Oberpfaffenhofen, Germany
[5]Geodesy and Glaciology, Bavarian Academy of Sciences and Humanities, Munich, Germany
[6]Institut für Geographie, Friedrich-Alexander-Universität Erlangen-Nuremberg, Erlangen, Germany

**Correspondence:** Anna Wendleder (anna.wendleder@dlr.de)

**Abstract.** Glacial surface meltwater directly influences glacier dynamics. However, in the case of debris-covered glaciers, the drivers of glacier velocity and the influence of supraglacial lakes have not yet been sufficiently analyzed and understood. We present a spatio-temporal analysis of key glacier characteristics for the Baltoro Glacier in the Karakoram, from October 2016 to September 2022 based on Earth Observation data and climate parameters extracted from the High Asia Refined analysis (HAR) data set. For the glacier variables, we used surface velocity, supraglacial lake extent, melt of snow and ice, and proglacial runoff-index derived from Earth Observation data. For climate variables, we focused on air temperature and precipitation. The surface velocity of the Baltoro Glacier was characterized by a spring speed-up, summer peak, and fall speed-up with a relative increase in summer by 0.2-0.3 m d$^{-1}$ (75-100%) in relation to winter velocities triggered by the onset or an increase of basal sliding. Snow and ice melt has the largest impact to the spring speed-up, summer velocity peak, and to the transition from inefficient to efficient subglacial drainage. It covered up to 64% (353 km$^2$) of the complete (debris-covered and debris-free) Baltoro Glacier and reached up to 4700 m a.s.l. during the first melt peak and up to 5600 m a.s.l. during summer. The temporal delay between the initial peak of seasonal melt and the first relative velocity maximum decreases downglacier. Drainage from supraglacial lakes (3.6-5.9 km$^2$) contributed to the fall speed-up, which showed a lower magnitude by 0.1-0.2 m d$^{-1}$ (20-30%) than the summer velocity peak. Most of the runoff can be attributed to the melt of snow and ice. However from mid-June onwards, the lakes play an increasing role, even though their contribution is estimated to be only about half of that of the melt. The observed increase in summer air temperatures leads to a greater extent of melt, as well as to a rise in number and total area of supraglacial lakes. This tendency is expected to intensify in a future warming climate.

## 1 Introduction

Glacial meltwater is an important control over glacier dynamics. In particular, the seasonal evolution and variation of glacier flow is strongly influenced by the timing and amount of meltwater (Glasser, 2013; Iken and Bindschadler, 1986). With the melt

onset, surface meltwater is formed and possibly drains to the glacier bed via crevasses or englacial pathways and is stored in cavities and distributed channels (Röthlisberger, 1972; Cuffey and Paterson, 2010). This influx of meltwater into the subglacial drainage system leads to an increase in basal water pressure. When subglacial water pressure approaches ice-overburden pressure, basal traction decreases and sliding increases as the ice decouples from the bed. The glacier - in case of a hard glacier bed - lifts up and accelerates (Weertman, 1964; Lliboutry, 1968; Iken and Bindschadler, 1986; Nolan and Echelmeyer, 1999; Sugiyama et al., 2011; Hoffman et al., 2016; Benn et al., 2019). Additionally, the inflow of water through small and incipient subglacial channels generates frictional heat which melts the ice walls and thus expands the channels leading ultimately to the formation of an efficient drainage system (Röthlisberger, 1972; Flowers, 2015). In an efficient subglacial channel system, water storage is reduced and water pressure in the channels lessens and glacier flow velocity decreases (Benn et al., 2019). In the absence of meltwater the effective pressure is larger and leads to a closure of the channels through regelation (Cuffey and Paterson, 2010) and creep (Benn et al., 2019; Flowers, 2015; Jiskoot, 2011).

Glacier movement is governed by gravitational driving stress and balanced by the resistive stresses. The driving stress is influenced by gravitational acceleration, ice density, thicknesses and surface slope and varies slowly. The resistive stresses arise from the drag at the glacier and by dynamical flow resistance. Glacier flow processes can be divided into ice deformation, basal sliding, and sediment deformation. Sliding and bed deformation occur only in the case of temperate and polythermal glaciers with a higher contribution of basal sliding than internal deformation velocities (Boulton and Hindmarsh, 1987; Jiskoot, 2011).

Glacier speed-ups in spring (Mair et al., 2001; Macgregor et al., 2005; Nanni et al., 2023), summer (Iken and Bindschadler, 1986; Copland et al., 2003; Bartholomaus et al., 2008; Quincey et al., 2009; Hewitt, 2013; Werder et al., 2013; Van Wychen et al., 2014; Armstrong et al., 2017; Nanni et al., 2023; Rada Giacaman and Schoof, 2023) and winter (Burgess et al., 2013; Hart et al., 2022) have been widely observed and their changes linked to an increase of air temperature, surface melt, and subglacial hydrology. In the case of debris-free glaciers, meltwater creation can be directly associated with warmer air temperatures and surface melt. Debris-covered glaciers, however, have a more complex, non-linear ablation with enhanced melting in areas of thin debris cover (few centimeters), thermal insulation in areas of thick debris coverage (Østrem, 1959; Nicholson and Benn, 2006), and melting hot-spots at ice cliffs (Brun et al., 2018; Buri et al., 2021) and supraglacial lakes (Miles et al., 2020). In an inefficient subglacial drainage system, supraglacial lake discharge can support basal sliding and hence cause higher glacier velocities (Sakai and Fujita, 2006; Sakai, 2012; Watson et al., 2016; Benn et al., 2017; Miles et al., 2020). A few studies have observed that supraglacial lake drainage could lead to a transition from an inefficient to an efficient subglacial drainage system and hence lead to a slowdown of the glacier velocity (Vincent and Moreau, 2016; Stevens et al., 2022).

Previously, we presented a time series of annual and seasonal glacier surface velocities derived from multi-mission Synthetic Aperture Radar (SAR) data for Baltoro Glacier, located in the Karakoram, Pakistan, from 1992 to 2017 (Quincey et al., 2009; Wendleder et al., 2018). We could show that in some years, the acceleration lasted longer and affected a larger glacier area than in others. In years with higher velocities, the supraglacial lakes mapped from Landsat and ASTER imagery were characterized by a larger number and a larger total area as well. However, only one image for each summer was available for mapping and did not provide sufficient insight into the seasonal evolution of the supraglacial lakes or their link to surface velocity. Therefore, we developed an approach using multi-temporal and multi-sensor Earth Observation data to provide a dense, almost daily summer

time series of supraglacial lakes (Wendleder et al., 2021a). Nevertheless, there is still a lack of detailed process understanding of whether and how the development of supraglacial lakes triggers basal sliding.

In this study, we combined relevant glacier variables derived from Earth Observation data and climate records to assess the extent to which the Baltoro Glacier responds, spatially or temporally, to a given climatic forcing. The processes and relationship of the variables were examined by statistical analysis. For the glacier variables, we used 1) surface velocity derived by intensity offset tracking from Sentinel-1 time series, 2) supraglacial lakes mapped by a random forest classifier applied on Sentinel-2, PlanetScope, Sentinel-1, and TerraSAR-X data, 3) melt of snow and ice detected using a change detection algorithm based on Sentinel-1, and 4) runoff-index estimated as surface areal coverage of the proglacial stream given in km$^2$ used as a proxy of the quantitative runoff from Sentinel-2 and PlanetScope imagery. For the climate variables we focussed on air temperature and precipitation extracted from the High Asia Refined analysis (HAR) data set. The analysis focused on the period from October 2016 to September 2022 providing a dense and continuous time series.

## 2  Study site

The Baltoro Glacier is located in the eastern Karakoram in the northern part of Pakistan (Figure 1). The glacier has a length of about 63 km and, together with its tributary glaciers, covers an area of approximately 554 km$^2$. Above Concordia (4600 m a.s.l.), the two major tributaries Godwin Austin Glacier and Baltoro South Glacier converge to the main Baltoro Glacier and change their flow direction westward. Several major tributary glaciers merge with the main branch along its northern and southern margins (Mayer et al., 2006; Quincey et al., 2009). Surface velocities range between 0.5 to 0.65 m d$^{-1}$ in summer and 0.3 to 0.4 m d$^{-1}$ in winter between Concordia and Urdukas (3900 m a.s.l.), and decrease to 0.03 to 0.1 m d$^{-1}$ near the terminus at 3400 m a.s.l. (Quincey et al., 2009; Wendleder et al., 2018). Approximately 38 % of the Baltoro Glacier is debris covered with a thin layer of 5-15 cm at Concordia, 30-40 cm at Urdukas, and in the order of 1 m thickness near the glacier terminus (Mayer et al., 2006; Quincey et al., 2009). Furthermore, the debris thickness varies spatially due to advection, debris and meltwater movement, and slow cycles of topographic inversion (Nicholson et al., 2018; Huo et al., 2021) which impedes the determination of the area proportion enhancing surface melt. Debris-covered glaciers are characterized by the presence of ice cliffs (Mayer et al., 2006; Evatt et al., 2017) and supraglacial lakes (Wendleder et al., 2021a). On the Baltoro Glacier ice cliffs are found between the terminus and Gore. Supraglacial lakes are located on the main glacier from the terminus up to Concordia. They usually fill between mid-April to mid-June and drain between mid-June to mid-September.

The Karakoram has a mid-latitude high-mountain climate with cold winters and mild summers. The regional climate is influenced by winter westerly disturbances with dominant winter and spring snowfall (Mayer et al., 2014; Dobreva et al., 2017), Indian summer monsoon with higher liquid precipitation, temperatures, and cloud coverage (Thayyen and Gergan, 2010; Bookhagen and Burbank, 2006), and the predominantly stable Tibetan Anticyclone. In the case of an irregular weakening of the Tibetan Anticyclone and thus causing an incursion of the Indian summer monsoon, large amounts of summer precipitation can be observed (Dobreva et al., 2017). The climate variables are strongly determined by the topography. Hence, precipitation increases with altitude reaching a mean annual rate of approximately 1600 mm pear year at 5300 m a.s.l. (Godwin Austen

region) and at 5500 m a.s.l. (Baltoro South region). The average daytime air temperature during summer is close to the freezing

point at 5400 m a.s.l., consequently most of the precipitation deposits as snow above this elevation (Mayer et al., 2006).

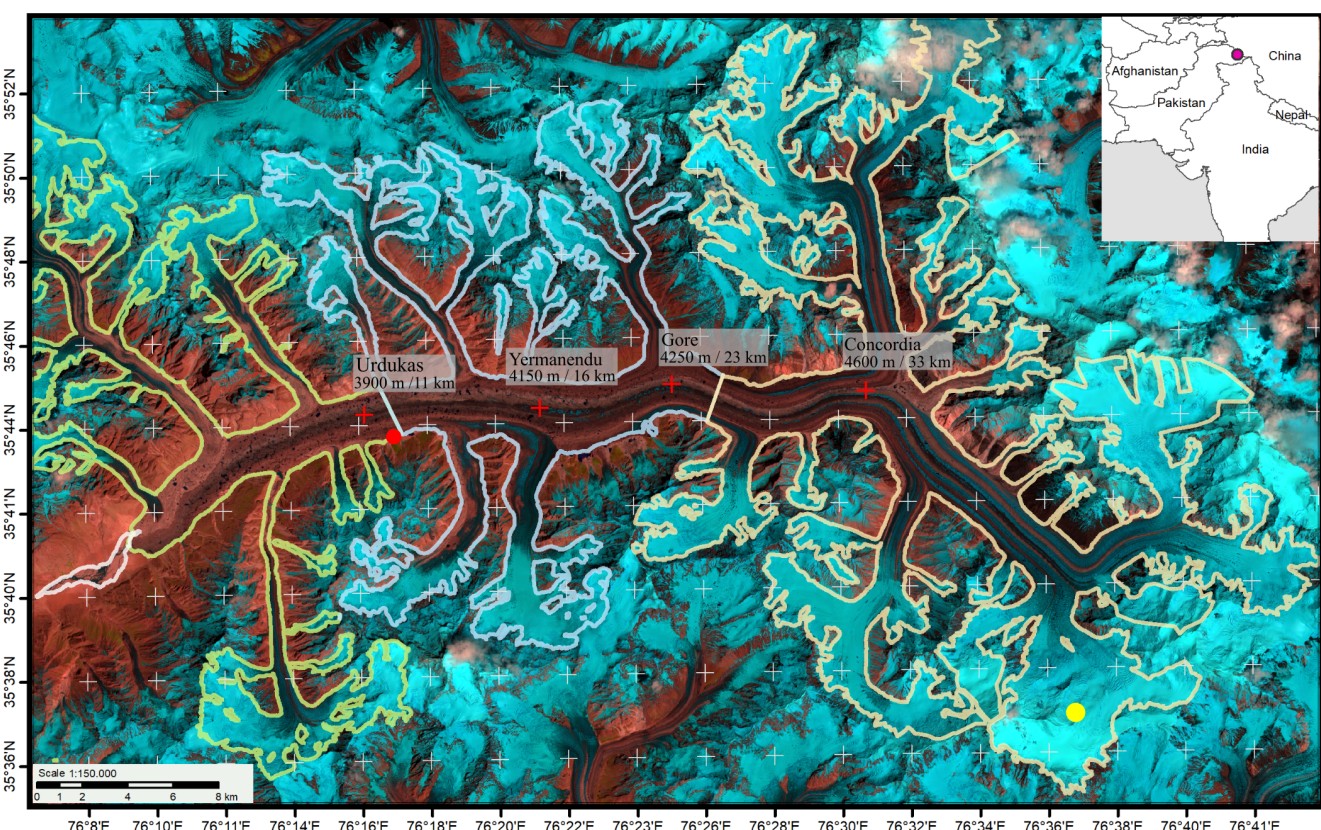

**Figure 1.** Overview of the Baltoro Glacier, Pakistan. Important place names are indicated. Red crosses marks the locations where the glacier surface velocity was extracted. The red point indicates the location of Urdukas, which differs by 3 km from the extracted location for the glacier velocity. The area of the mapped runoff is displayed with a white polygon. The western, central, and eastern sector for the mapping of the snow and ice melt is displayed in yellow, blue, and orange, respectively. The air temperature was extracted at Urdukas and the precipitation at Baltoro South Glacier which is marked with a yellow point (center of the HAR pixel). The background image is a Sentinel-2 composite (shortwave-infrared, near-infrared, red) acquired on 22 July 2019.

## 3 Material and Methods

This section describes the used Earth Observation data, known methods to process them into level-2 data, and the used reanalysis data. Furthermore, the analyses of the spatio-temporal relationship of the variables, the Pearson correlation and the linear regression to quantify the direction and strength of the relationship and the dependency are explained.

### 3.1 Glacier surface velocity

The glacier surface velocity was calculated from the Sentinel-1 Interferometric Wide Swath Single Look Complex (SLC) data with a spatial resolution of 10 m. We used only data from the ascending orbit as only these were continuously available for the complete observation period. The intensity offset tracking algorithm was applied to consecutive pairs of co-registered SAR intensity images (Strozzi et al., 2002; Friedl et al., 2018). Feature tracking and subsequent processing steps were implemented in GAMMA (release 20211201, (Wegmüller et al., 2016)). Tracking patch sizes and step sizes were adapted to sensor specifications and expected displacement lying at 250 x 50 pixel and 50 x 10 pixel, respectively. The Copernicus Digital Elevation Model (DEM) (30 m, version 2022_1, (ESA, 2019)) provided the topographic reference for geocoding and orthorectification of the surface velocity maps. The mean accuracy of the velocity maps is 0.06 m d$^{-1}$ resulting in a standard deviation of 0.042 m d$^{-1}$ which is calculated according to Strozzi et al. (2002) and Friedl et al. (2021). For the analysis, we extracted the surface velocity values along the glacier centerline at four different locations (Figure 1), namely at Urdukas (11 km), at the confluence with Yermanendu Glacier (16 km), Gore (23 km), and Concordia (33 km distance from terminus). The four different points were selected to best reflect the spatial variation along the glacier.

### 3.2 Supraglacial lake mapping

Supraglacial lakes were mapped based on a multi-sensor and multi-temporal summer time series from 2016 to 2022 acquired by the optical sensors Sentinel-2 and PlanetScope as well as the SAR sensors Sentinel-1 and TerraSAR-X. The Sentinel-2 Multi Spectral Instrument (MSI) orthorectified Level-1C Top-Of-Atmospheric products (spatial resolution of 10 m) were atmospherically corrected to L2A products using MAJA (MACCS ATCOR Joint Algorithm, release 4.2, (Hagolle et al., 2017)). The PlanetScope Analytic Ortho Scene Products (Level 3B, spatial resolution of 3 m) were downloaded as orthorectified and atmospherically corrected Surface Reflectance (SR) data. In our approach, a coregistration of the PlanetScope data is not needed since the assignment of the classified lakes over the season is performed using their center coordinate. Sentinel-1 Interferometric Wide Swath Single Look Complex (SLC, spatial resolution of 10 m) C-band and TerraSAR-X ScanSAR Multi-Look Ground Range Detected (MGD, spatial resolution of 40 m) X-band data were processed to Analysis Ready Data (ARD) using the Multi-SAR System (Schmitt et al., 2015, 2020). The mapping of the seasonal lake evolution used a semi-automatic approach which is based on a random forest classifier applied separately to each sensor. To produce a consistent and internally robust time series, a combination of linear regression and the Hausdorff distance (Hausdorff, 1914) were used to harmonize SAR- and optical-derived lake areas. The advantage of the multi-sensor approach is that it highlights the strengths of each sensor and compensates for their weaknesses: Sentinel-2 provides a continuous, radiometrically stable time series with

a temporal resolution of 12 days, which is filled by the high temporal sampling of PlanetScope data. During periods of cloud cover SAR data provide important information. The disadvantages of the SAR data, i.e. lake area underestimation and missing data from side-looking radar geometry and undulating glacier surface topographies, can be compensated by using optical data acquired on the same day. The time series has a temporal sampling of 2-4 days with a spatial sampling of 10 m. The mean relative Root Squared Error is at 1.0 % (total area of 9.151 km$^2$ with an absolute RSE of 0.0945 km$^2$). Detailed processing steps as well as distribution and life span of the supraglacial lakes for the years 2016 to 2020 were published by Wendleder et al. (2021b). For this study, the time series was extended to 2021 and 2022. The filling and draining of the supraglacial lakes is parameterized by their aggregated area.

### 3.3 Glacier runoff mapping

Since we only had Earth Observation data available, we were limited to mapping the width of the glacier discharge stream and using the estimated surface areal coverage of the proglacial stream as proxy of the quantitative runoff given in km$^2$. To avoid misunderstanding to conventional runoff measurements given in volume, we are using the term 'runoff-index'. The mapping was only applied to the area of the upper proglacial stream (2.7 km$^2$ area and 5.6 km length) bordered to south and north by alluvial fans and slopes (see white polygon in Figure 1). From the low-flow to peak discharge period, the water-filled channel width at the glacier's terminus increased tenfold from 70 m to around 700 m. We used the Sentinel-2 Multi Spectral Instrument (MSI) orthorectified and MAJA atmospherically corrected L2A products (spatial resolution of 10 m) and the PlanetScope Analytic Ortho Scene Products (Level 3B) Surface Reflectance (SR) data (spatial resolution of 3 m). Due to the high turbidity of the glacier discharge, water and sandy soil are hard to differentiate using the Normalized Different Water Index. Therefore, the optical data of each sensor were first fused to Kennaugh Elements using the red, blue, and near-infrared bands. In radar, the Kennaugh Elements describes the polarimetric information and enables the interpretation of physical scattering mechanisms. In the case of optical data, the Kennaugh Elements are a fusion technique to combine multiple multi-spectral bands to one image while enhancing their spectral characteristics (Schmitt et al., 2020). Afterwards, a K-means clustering grouped each image into the two classes "runoff" and "background". The class "runoff" was assigned by a fixed, manually selected point on the western border of the mapping area that was always covered with water. To evaluate the accuracy, four reference data sets with different acquisition dates of the run-off area were digitized manually on the basis of the near-infrared band. The classification achieved a mean relative RSE of 0.12 % (total area of 3.1 km$^2$ with an absolute RSE of 0.14 km$^2$). The time series has a temporal sampling of 5-15 days in the months of May and June, which is characterized by a higher cloud coverage and 1-5 days in the months of July to September with less cloud coverage.

### 3.4 Mapping of snow and ice melt

We mapped the snow and ice melt on the complete Baltoro Glacier (see outline of the three polygons in Figure 1) which includes the debris-covered and debris-free part of the glacier from the Sentinel-1 Interferometric Wide Swath C-band data with a spatial resolution of 10 m. The processing of the SLC to ARD data was performed with the Multi-SAR System (Schmitt et al., 2020). The additional use of image enhancement (specifically, multi-scale multi-looking) resulted in a smoothing of

noise and hence a more homogeneous environment and a better classification (Schmitt et al., 2015; Wendleder et al., 2022). As cross-polarisation (VH) has a greater absorption over wet surfaces than the VV-polarization, leading to a lower backscatter and hence a better discrimination of wet surfaces (Rott and Mätzler, 1987), we used only this polarization. For every scene, the image difference to a reference scene was calculated. To ensure cold air temperatures prevailed during the acquisition, the first scene in January of each year, the coldest month, was chosen as the reference scene. As threshold we used one half of the signal power (-3 dB) (Shi and Dozier, 1995; Nagler and Rott, 2000; Scher et al., 2021). This approach achieves an overall accuracy of 81.4%-85.4% (Nagler and Rott, 2000). By intersecting the mapping of the melt with the Copernicus DEM (1 arc second, version 2022_1, (ESA, 2019)), the 10% and 90% percentile of the elevation of the aggregated melt area have been used, which changes over time. We selected only SAR acquisitions from the ascending orbit. Firstly, the ascending images were acquired during the afternoon (13:00 UTC, 18:00 local time) to better represent the maximum melt area than the descending images acquired during the morning (1:00 h UTC, 6:00h local time). Secondly, the accumulation areas of the tributary glaciers were imaged with only minor influence of layover or radar shadow as the slopes faced away from the sensor (Wendleder et al., 2021a). The mapping has a temporal resolution of 12 days. For the mapping, we divided the glacier into the western (from glacier terminus to Urdukas), central (from Urdukas to Gore), and eastern area (upwards of Gore). As area and elevation of the glacier increases from west to east, the temporal delay of the melt in the three sectors reflects the vertical gradient.

## 3.5 Air Temperature and Precipitation

The near surface air temperature at 12 pm (noon, local time) and total precipitation data (daily sum) were obtained from the daily interpolated HAR data set (version 2) provided by the Chair of Climatology, TU Berlin. The air temperature data (2 m height) from an automatic weather station located at Urdukas (Mihalcea et al., 2008) was needed for the local downscaling of the HAR data. The weather station is equipped with an air temperature sensor (thermo-hygrometer with radiation shield) taking hourly measurements in 2011. The global European Centre for Medium-Range Weather Forecasts (ECMWF) Re-Analysis (ERA-5, spatial resolution of 80 km) was dynamically downscaled by regional climate models for the High-Mountain Asia to produce the regional refined HAR data set with a spatial resolution of 10 km. HAR is available for the period 1980 to 2022 (Wang et al., 2021). Compared with in-situ data, the modelled air temperatures show a mean bias of -0.58 K (Wang et al., 2021). The elevation model used for the downscaling with its spatial resolution of 10 km is too coarse to represent the complex terrain of the Karakoram, which leads to an underestimation of air temperatures in the lower glacier and an overestimation of temperatures on the accumulation area. To downscale the air temperatures, a mean lapse rate for each month was determined based on air temperatures data from the automatic weather station using a fitting curve (3rd order). The derived lapse rates varied between 7° C km$^{-1}$ (December) and 11° C km$^{-1}$ (June). The precipitation was extracted at Baltoro South Glacier, one of the two major tributaries with the larger glacier area.

## 3.6 Temporal relationship

The first step is to analyse if there is a temporal relationship between the supraglacial lakes, melt of snow and ice, glacier velocity, runoff-index, air temperature, and precipitation. For the presentation, we plotted all variables in one figure using the hydrological year (1 October-30 September) as time reference.

## 3.7 Spatial relationship

If the temporal representation shows a relationship in time between the variables supraglacial lakes, melt of snow and ice, and glacier velocity, the spatial representation demonstrates if the variables are spatially related as well, i.e. whether the location of drainage and velocity are nearby. Therefore, glacier velocity was displayed on a hexagonal grid known as H3 (Sahr, 2011; Brodsky, 2018; Fichtner et al., 2023). The advantage of a hexagonal compared to a rectangular grid is the simpler and more symmetric nearest neighborhood and the better clarity in visualizations. The conversion from the rectangle to hexagonal presentation is based on resampling using the median value. Only the supraglacial lakes that arose or drained in the observed period of higher glacier velocity were displayed in order to analyze if the draining water could be a control on ice dynamics.

## 3.8 Pearson Correlation

The glacier surface velocities were correlated with the supraglacial lake area, melt extent, runoff-index, and meteorological parameters. As each variable had a different temporal resolution, the time series were resampled to daily values. The Pearson correlation coefficient (r) was applied each year for two different periods: 1) the first period was defined from the day with the first positive air temperatures at Urdukas until the time of maximum glacier velocity and 2) the second period from the time of maximum glacier velocity until the end of the melt season on 30 September (Berthier and Brun, 2019). The significance of the correlation coefficients with a confidence level of 95% was estimated using the student distribution (Obilor and Amadi, 2018).

## 3.9 Linear Regression

To analyze the relationship between the datasets as well as the dependency of all variables for both periods separately, we applied a linear regression based on a least-square robust adjustment using $R^2$ and the regression coefficient (slope) to show the dependency of the variables. The significance of any observed trends was estimated using the Mann–Kendall test with a confidence level of 95%.

## 4 Results

First, we present the results of the spatio-temporal relationship, followed by the results of the Pearson correlation, linear regression, and the estimated temporal delay between melt and surface velocity.

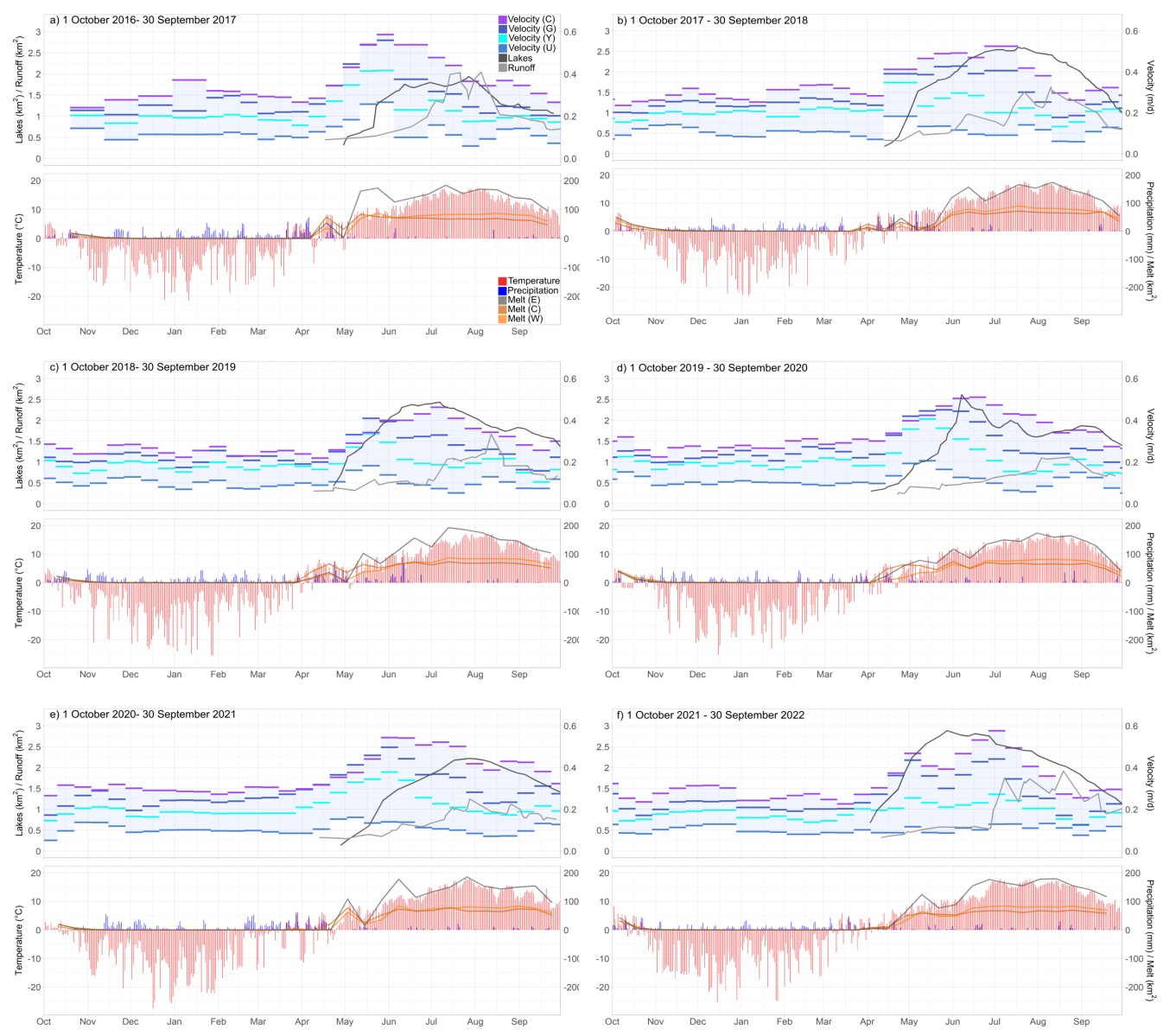

**Figure 2.** Temporal relationship of the supraglacial lakes, runoff-index, surface velocity, air temperature, precipitation, and melt for 1 October 2016 to 30 September 2022 (a-f). All images have the same legend shown in a). The label of the left y-axis is displayed on the outer left side, that of the right y-axis on the outer right side. The abbreviation U stands for Urdukas, Y for Yermanendu, G for Gore, and C for Concordia (surface velocity) whereas W stand for western, C for central, and E for eastern sector (melt).

Table 1. Glacier and climate variables with their relevant parameters for October 2016 to September 2022. The maximum area refers to the area at a specific point in time and the total summer area to the summarized area during summer. The abbreviation W stand for western, C for central, and E for eastern sector.

| | Oct 2016–Sept 2017 | Oct 2017–Sept 2018 | Oct 2018–Sept 2019 | Oct 2019–Sept 2020 | Oct 2020–Sept 2021 | Oct 2021–Sept 2022 |
|---|---|---|---|---|---|---|
| **Surface velocity** | | | | | | |
| Winter velocity (m d$^{-1}$) | 0.09-0.37 | 0.08-0.33 | 0.07-0.29 | 0.08-0.33 | 0.05-0.32 | 0.08-0.32 |
| Period of winter velocity | 20 Oct-17 April | 1 Oct-12 April | 1 Oct-12 April | 1 Oct-12 April | 1 Oct-12 April | 1 Oct-12 April |
| Spring speed-up velocity (m d$^{-1}$) | 0.15-0.54 | 0.34-0.41 | 0.41 | 0.45 | 0.46 | 0.47 |
| Period of spring speed-up | 18 April-23 May | 13 April-7 May | 2 May-25 May | 26 April-19 May | 21 April-26 May | 28 April-9 May |
| Peak velocity (m d$^{-1}$) | 0.59 | 0.53 | 0.46 | 0.51 | 0.54 | 0.58 |
| Period of peak velocity | 24 May-4 June | 26 June-18 July | 1 July-12 July | 13 June-24 June | 27 May-19 June | 27 June-8 July. |
| Fall speed-up velocity (m d$^{-1}$) | 0.37 | 0.32 | 0.33 | 0.34 | 0.43 | 0.32 |
| Period of fall speed-up velocity | 16 Aug-27 Aug | 4 Sept-27 Sept | 23 Sept-16 Oct | 5 Sept-16 Sept | 19 Aug-30 Sept | 19 Sept-30 Sept |
| **Supraglacial lake** | | | | | | |
| Number of lakes | 412 | 501 | 498 | 379 | 372 | 710 |
| Maximum area (km$^2$) | 1.9 | 2.6 | 2.4 | 2.0 | 2.5 | 2.9 |
| Total summer area (km$^2$) | 3.6 | 5.8 | 4.7 | 4.6 | 4.8 | 5.9 |
| Date of peak | 28 July | 23 July | 8 July | 12 July | 11 Aug | 28 May |
| **Runoff-index** | | | | | | |
| Date of peak (area (km$^2$)) | 21 July (2.1); 5 Aug (2.0) | 12 June (1.0); 22 July (1.5); 10 Aug (1.6) | 22 May (0.6); 16 July (1.1); 12 Aug (1.7) | 3 Aug (1.1); 25 Aug (1.1) | 10 June (0.6); 28 July (1.2); 26 Aug (1.1) | 16 June (0.6); 8 July (1.8); 26 July (1.7); 19 Aug (1.9) |
| Total summer area (km$^2$) | 36.0 | 32.0 | 27.7 | 17.0 | 31.4 | 27.3 |
| **Melt** | | | | | | |
| Period | 18 April-26 Dec | 1 April-22 Oct | 27 Marc-28 Dec | 14 April-16 Nov | 21 April-17 Dec | 4 April-24 Dec |
| Maximum area (km$^2$) | 336 | 328 | 353 | 319 | 335 | 326 |
| Max. area per W/C/E sector (km$^2$) | 87/84/184 | 90/72/173 | 87/73/193 | 81/74/174 | 85/79/185 | 84/69/179 |
| Total summer area (km$^2$) | 3697 | 2989 | 3697 | 3701 | 3551 | 3773 |
| **Air Temperature** | | | | | | |
| Degree day sum (°C) | 1989 | 1890 | 1822 | 1851 | 1911 | 2084 |
| Period with pos. air temperatures | 21 March–22 Oct | 12 Marc-1 Nov | 27 March-30 Oct | 21 March-28 Oct | 7 March-22 Oct | 05 March-5 Nov |
| **Precipitation** | | | | | | |
| Winter period (mm) | 3664 | 2621 | 3514 | 3128 | 3358 | 2670 |
| Summer period (mm) | 1468 | 1658 | 1754 | 1234 | 1088 | 914 |

## 4.1 Temporal relationship

The temporal relationship of the supraglacial lakes, runoff-index, surface velocity, air temperature, precipitation, and melt for October 2016 to September 2022 is shown in Figure 2. Table 1 lists the glacier and climate variables and their relevant parameters. The following subsections focus on selected seasonal effects and only consider the influencing parameters.

### 4.1.1 Warm spring season

During March (earliest: 5 March 2022, latest: 27 March 2019), daily noon air temperatures consistently rose into the positive range for the first time. In mid-April there was usually a period of higher air temperatures, coinciding with a preceding precipitation event. At this time, the melt line and the zero degree level (estimated from the air temperatures at Urdukas) was at 4700 m a.s.l., hence the precipitation above that altitude fell as snow. Afterwards, melt area and air temperatures exhibited a linear relationship which indicated that higher air temperatures were present in higher altitudes.

At the same time, the snow melted and supraglacial lakes formed. The melt started between 27 March (earliest; 2019) and 21 April (latest; 2021) and terminated at the end of October. First the snow melted in the lower altitudes (3600-4300 m a.s.l.) on the debris-covered main branch and debris-free tributary glaciers, later only on the debris-free glaciers in the higher altitudes (up to 4700 m a.s.l.).

The debris cover leads to heterogeneous ablation on the glacier surface with enhanced melting during warm periods on the ice cliffs and the supraglacial lakes. The formation of the lakes started early-April to early-May and reached the peak of the maximum total area from end-May/early-June (2020, 2022) through mid/end-July (2017, 2018, 2019) to mid-August (2021). Possibly the surface meltwater could initiate lake formation, though, it could not be proven due to the low spatial resolution of the remote sensing data. Spring 2022 was significantly affected by heat waves (Otto et al., 2023). In summer 2022, the degree day sum at Urdukas was at 2084 compared to value range of 1822 (2019) and 1989 (2017) which led to an early supraglacial lake peak (28 May).

### 4.1.2 Spring and summer speed-up

Between May and September, the melt of snow and ice fluctuated gradually through time between minimum and maximum extent, reaching spatial maxima of 90 km$^2$ for the western sector (2018), 84 km$^2$ for the central sector (2017), and 190 km$^2$ for the eastern sector (2019). In July 2019, the total melt area had a maximum of 353 km$^2$ and also covered the debris-free part of the glacier. It corresponds to 64% of the total area of Baltoro Glacier. A temporal delay from the lower (western) sector to the upper (eastern) sector was hardly discernible which could probably be explained by an abrupt transition from winter to spring with higher air temperatures at higher altitudes. Differences only existed in the melt area as the east sector covers a larger glacier area than the western and central sector. The first and second melt event occurred in the same period as the spring speed-up and lake evolution. In the area upwards of Urdukas, which is characterized by less debris cover, the meltwater accumulates in the supraglacial lakes and drains into the subglacial system or flows laterally into larger streams. In the area

downstream of Urdukas, there are no larger supraglacial channels and the water can drain through moulins and crevasses to the ground.

The winter surface velocities (1 October until the first date with positive air temperatures usually at the beginning/end of March) ranged over the years between 0.05 and 0.37 m d$^{-1}$ and were followed by a spring speed-up. This event lasted usually from the end of April to the beginning of May and led to a relative increase of 0.1–0.16 m d$^{-1}$ (30-35%) above Yermanendu. In 2017, 2019, 2020, and 2021, the velocities at Gore exceeded even those at Concordia at this time. The spring speed-up was particularly visible in 2018 affecting all four locations. However, in 2022, the velocities at Urdukas were rarely impacted.

After the spring speed-up, the flow at Concordia accelerated continuously until the summer peak of 0.46-0.59 m d$^{-1}$ was reached in June to July. Only in 2022, a short slow-down between the speed-up and the summer peak was recognizable. The high lasted for 11-12 days in 2017, 2019, 2021, and 2022 and 22 days in 2018 and 2020. However, not all four locations experienced the maximum velocity at the same time: in 2022, the summer peak affected all four locations; in 2021, only the area upglacier of Yermanendu was influenced; in 2020 and 2019, the velocities downglacier of Gore reached their maximum during the spring speed-up; in 2018, the velocities at Concordia and Gore had their high during summer peak, Yermanendu and Urdukas during spring speed-up; in 2017, the velocities upwards of Yermanendu reached their maximum during summer peak and Urdukas afterwards. Additionally, the year 2021 was characterized by a continuous transition of spring speed-up and summer peak. Between June and July, the surface velocities between Concordia and Urdukas spread with a difference of up to 0.46 m d$^{-1}$ with an increase at Concordia and decrease at Urdukas. In 2017 and 2021, this phenomenon happened after the summer peak and in 2018-2020 between speed-up and summer peak.

### 4.1.3  Fall speed-up

Supraglacial lakes store large quantities of water and can hence influence the velocity pattern in the event of rapid drainage. The maximum area of the supraglacial lakes ranged between 1.9 km$^2$ (2019) and 3.1 km$^2$ (2022). The drainage started early-June to mid-July. In 2018 to 2020, it occurred in the period around the summer velocity peak and coincided with a velocity decrease. Though, in 2017 and 2021, lake drainage started shortly before the fall speed-up and in 2022, between spring speed-up and summer peak. The supraglacial lakes drained after the glacier slowdown from mid-August to mid-September and lead to a velocity acceleration from the winter velocities to 0.35-0.5 m d$^{-1}$ at Concordia in mid-September.

### 4.1.4  Runoff

The runoff intensity is an indicator of when the water began to flow continuously and thus marks the transition from inefficient to efficient subglacial drainage. It rose from July onwards after the maximum of summer speed-up and supraglacial lake and during the melt. The pattern and peak of the runoff-index changed every year: 2017 was characterized with a slow increase during May and June with two distinctive peaks in mid-July (21 July) and early-August (5 August); 2018 by a steady and slow rise with lows and highs (12 June, 22 July, 10 August); 2019 by a significant increase beginning of July, a constant run-off until end of July, and a subsequent peak in early August; 2020 by a slow gain from May until August and two peaks (3 and 25 August) with continuous runoff-index between both dates; 2021 by a continuous increase until end of July succeeded by a

high (28 July); 2022 by a continuous low runoff-index until the end of June and a sudden rise and thus to a distinct transition to an efficient subglacial drainage.

## 4.2 Spatial relationship

The spatial relationship of supraglacial lakes, melt, and surface velocity for the two periods of spring speed-up (13 April-07 May) and summer peak (07 May-18 July) is presented as an example for 2018 in Figure 3. This year was chosen because the events were clearly recognizable. In the first period (Fig. 3a), the surface velocity between Urdukas and Gore ranged from 0.2 to 0.4 m d$^{-1}$ and between Gore and Concordia from 0.4-0.6 m d$^{-1}$. During this time, the main branch between Gore and Concordia and the neighboring tributary glaciers up to 4700 m a.s.l. exhibited snow and ice melt covering an area of 237 km$^2$ (Figure 3a

displayed in orange). The supraglacial lakes first formed between the glacier tongue and Yermanendu (increase of 1.1 km$^2$). In the second period (Fig. 3b), the higher velocities of 0.4-0.6 m d$^{-1}$ expanded up to 5 km below Yermanendu and the melt upwards to 5600 m a.s.l. and covered all neighboring tributary glaciers with an area of 327 km$^2$ (Figure 3b displayed in orange). Though, the supraglacial lake formation extended up to Gore. During this period, the supraglacial lakes are characterized by a filling from 7 May to 2 July by 1.3 km$^2$, followed by filling (0.33 km$^2$) and simultaneous draining (0.46 km$^2$) from 2 July to

18 July.

## 4.3 Pearson Correlation

Figure 4 lists the Pearson correlation of all variables for the first period defined from the first day with positive air temperatures to the maximum glacier velocity and the second period defined from the maximum glacier velocity until the end of the ablation season. Correlation results can be found in the supplementary materials (Table A1 and A2)). In the two periods, the correlation

of the variables temperature and melt (0.7-0.95) as well as temperature and supraglacial lake (0.58-0.89) showed a similar pattern with a correlation above 0.58 though with negligible minor changes in the Pearson correlation during both periods (0.1-0.2); except for the decrease by 0.4 for temperatures and supraglacial lakes in the second period of 2020. A high degree of correlation existed between melt and runoff-index (0.65–0.93) and supraglacial lakes (0.63–0.94), respectively, albeit with outliers in both relationships in the second period of 2020 (decrease by 0.4 and 0.9). The relationship between supraglacial

lakes and velocity as well as melt and velocity showed the same pattern with higher correlations at Concordia and Yermanendu and lower correlations in Urdukas in both periods. However, the correlations of the second period vary significantly (0.5 to -0.8) over the years and generally decrease by 0.4-1.3 in the second period. Particularly noticeable was the variation between positive and negative correlation within the observation period for Urdukas in both parameters. The correlation between velocity and runoff decreased by 1.3 in the second period. However, compared to the two pairs of variables mentioned above, the second

period is characterised by a heterogeneous pattern over the years.

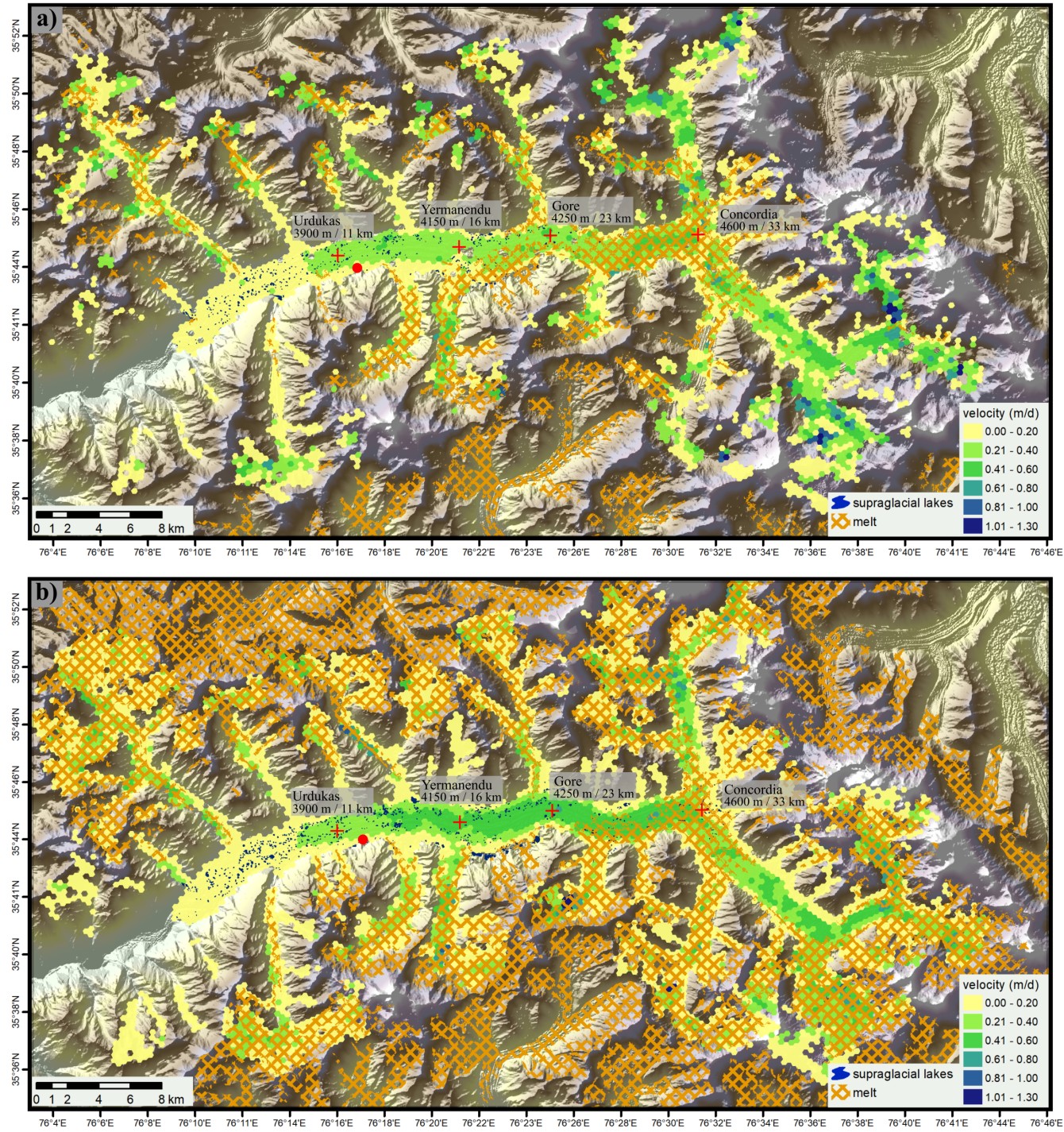

**Figure 3.** Spatial relationship of the maximum supraglacial lake area, aggregated melt, and median surface velocity from 13 April to 7 May 2018 (a) and 7 May to 18 July 2018 (b)

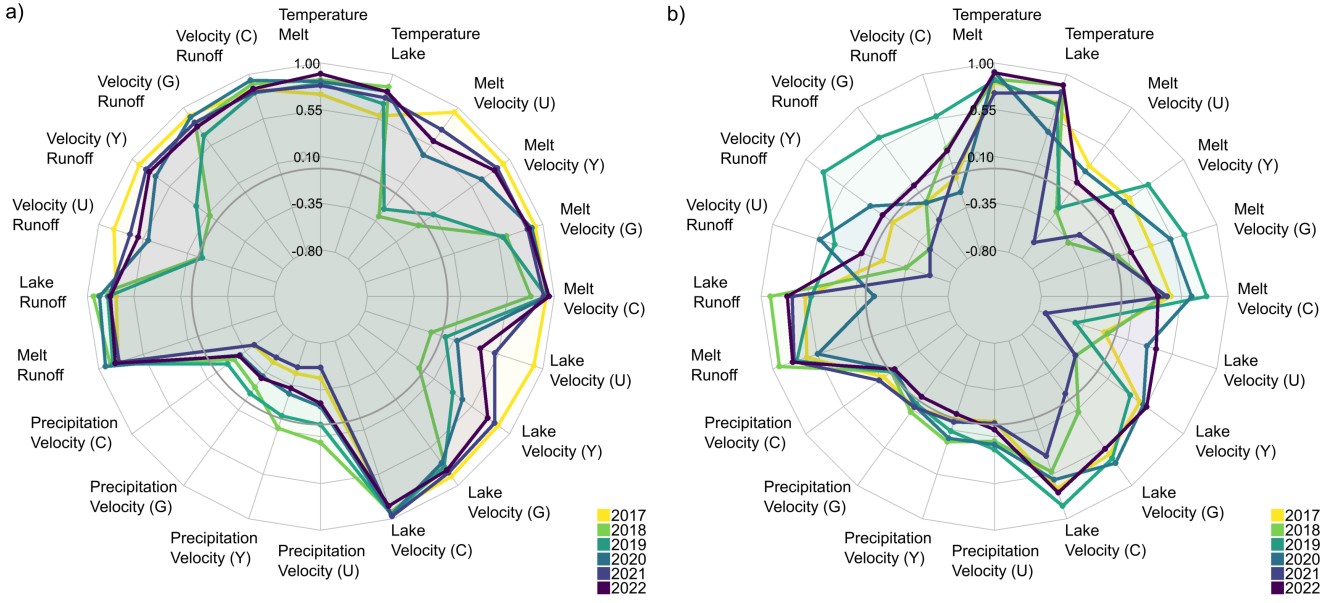

**Figure 4.** Pearson correlation of glacier and climate variables for first period defined from the first day with positive air temperatures to the maximum glacier velocity (a) and second period defined from the maximum glacier velocity until the end of the ablation season (b). The null line is displayed in grey. The abbreviation U stands for Urdukas, Y for Yermanendu, G for Gore, and C for Concordia.

### 4.4 Linear Regression

Table 2 and 3 list the results of the linear regression with $R^2$ and the regression coefficient. In 2018, 2020, and 2022, air temperature and melt as well as air temperature and supraglacial lakes have a high $R^2$ (0.65-0.83), but the dependency is low (0-7 °C/km$^2$). Melt and surface velocity have a high $R^2$ for the first period in 2017 on the area up-glacier of Urdukas (0.85-0.88), 2019 at Concordia (0.84), in 2020 (0.78-0.87) and 2022 (0.74-0.9) up-glacier of Gore, and in 2021 (0.7-0.8) up-glacier of Yermanendu. The regression coefficient lies at 832-1897 km$^2$d/m. The second period shows a low $R^2$. Supraglacial lakes and surface velocity shows a high correlation up-glacier of Urdukas in the first period of 2017 (0.73-0.87), up-glacier of Gore in the first period of 2021 (0.71-0.94) and at Concordia in the first period of 2018 (0.86), 2019 (0.92), 2020 (0.91), and 2022 (0.75) and in the second period of 2019 (0.75). Though, the dependency of supraglacial lakes to velocity is minimal (4-11 km$^2$d/m). $R^2$ and dependency of precipitation to surface velocity is not significant. Melt and runoff-index as well as supraglacial lakes and runoff-index show the pattern with similar $R^2$ (0.77-0.88), however the dependency of melt (171-370 km$^2$/km$^2$) is larger than the one of supraglacial lakes (1-4 km$^2$/km$^2$). The dependency of the surface velocity to runoff-index is not significant (0 m/d km$^2$).

**Table 2.** Linear regression given as $R^2$ and the regression coefficient (in parentheses) of all variables for 2017-2019. Only significant trends according to a 95% confidence interval based on the Mann–Kendall test are shown. Results with an $R^2$ smaller than 0.3 are displayed in red and those larger than 0.7 are displayed in green.

| | Period I 2017 21.03.-04.06. | Period II 2017 05.06.-30.09. | Period I 2018 12.03.-17.07. | Period II 2018 18.07.-30.09. | Period I 2019 27.03.-12.07. | Period II 2019 13.07.–30.09. |
|---|---|---|---|---|---|---|
| Air temperature – Melt (°C/km²) | 0.48 (0) | 0.65 (0) | 0.7 (0) | 0.7 (0) | 0.64 (0) | 0.68 (0) |
| Air temperature – Supraglacial lake (°C/km²) | 0.32 (5) | 0.5 (7) | 0.76 (5) | 0.78 (7) | 0.49 (3) | 0.47 (9) |
| Melt – Velocity at Urdukas (km²d/m) | 0.88 (1897) | 0.08 (455) | | | 0.03 (-557) | |
| Melt – Velocity at Yermanendu (km²d/m) | 0.84 (1161) | 0.11 (511) | | 0.12 (-598) | 0.0 (153) | 0.32 (742) |
| Melt – Velocity at Gore (km²d/m) | 0.85 (889) | 0.10 (235) | 0.39 (1053) | | 0.34 (754) | 0.44 (707) |
| Melt – Velocity at Concordia (km²d/m) | 0.85 (970) | 0.19 (225) | 0.59 (1061) | 0.12 (374) | 0.84 (1007) | 0.62 (783) |
| Supraglacial lake – Velocity at Urdukas (km²d/m) | 0.82 (8) | | 0.00 (-4) | | 0.0 (0) | |
| Supraglacial lake – Velocity at Yermanendu (km²d/m) | 0.73 (5) | 0.22 (4) | | | 0.0 (5) | |
| Supraglacial lake – Velocity at Gore (km²d/m) | 0.79 (4) | 0.39 (3) | 0.63 (11) | | 0.59 (10) | 0.45 (3) |
| Supraglacial lake – Velocity at Concordia (km²d/m) | 0.87 (4) | 0.47 (2) | 0.86 (11) | 0.26 (5) | 0.92 (11) | 0.75 (4) |
| Precipitation – Velocity at Urdukas (mm d/m) | 0.20 (-131) | 0.0 (-8) | | | | |
| Precipitation – Velocity at Yermanendu (mm d/m) | 0.21 (-84) | | | | | |
| Precipitation – Velocity at Gore (mm d/m) | 0.21 (-63) | 0.0 (5) | 0.02 (-31) | | 0.0 (-20) | |
| Precipitation – Velocity at Concordia (mm d/m) | 0.19 (-67) | 0.0 (4) | 0.04 (-33) | | 0.01 (-26) | -0.01 (-1) |
| Melt - Runoff-index (km²/km²) | 0.64 (411) | 0.41 (66) | 0.77 (295) | 0.86 (171) | 0.73 (370) | 0.56 (141) |
| Supraglacial lake - Runoff-index (km²/km²) | 0.51 (2) | 0.32 (0) | 0.87 (3) | 0.82 (1) | 0.64 (4) | 0.25 (0) |
| Velocity at Urdukas - Runoff-index (m/d km²) | 0.71 (0) | 0.00 (0) | | | | 0.13 (0) |
| Velocity at Yermanendu - Runoff-index (m/d km²) | 0.82 (0) | 0.0 (0) | | 0.21 (0) | 0.04 (0) | 0.61 (0) |
| Velocity at Gore - Runoff-index (m/d km²) | 0.80 (0) | | 0.64 (0) | 0 (0.0) | 0.43 (0) | 0 (0.4) |
| Velocity at Concordia - Runoff-index (m/d km²) | 0.71 (0) | | 0.80 (0) | | 0.65 (0) | 0.32 (0) |

## 4.5 Temporal delay between melt and velocity

Following the high correlation of melt area and surface velocity during the spring period almost every year and the process-oriented relation between melt onset, inefficient subglacial drainage and basal sliding, we investigated the temporal delay between first seasonal melt peak and first relative velocity maximum at Urdukas, Yermanendu, Gore, and Concordia which is shown in Fig. 5. The temporal lag between the two parameters is generally longest at Concordia ($34 \pm 25$ days), followed downglacier by Gore ($12 \pm 19$ days) and Yermanendu ($12 \pm 14$ days), and shortest at Urdukas ($10 \pm 16$ days). Given the period from 2017 to 2022 there is an indication that the temporal delay between the first seasonal melt peak and the first relative velocity maximum is decreasing. Thus the amount of meltwater causing glacier velocity variations is reached in a shorter which will be more supported by warmer climates in the future. As the calculation bases only on six years, this observation requires a careful interpretation and needs further investigation.

**Table 3.** Linear regression given as $R^2$ and the regression coefficient (in parentheses) of all variables for 2020-2022. Only significant trends according to a 95% confidence interval based on the Mann–Kendall test are shown. Results with an $R^2$ smaller than 0.3 are displayed in red and those larger than 0.7 are displayed in green.

| | Period I 2020 21.03.-24.06. | Period II 2020 25.06.-30.09. | Period I 2021 07.03.-19.06. | Period II 2021 20.06.-30.09. | Period I 2022 05.03.-08.07. | Period II 2022 09.07.–30.09. |
|---|---|---|---|---|---|---|
| Air temperature – Melt (°C/km$^2$) | 0.67 (0) | 0.83 (0) | 0.61 (0) | 0.5 (0) | 0.80 (0) | 0.81 (0) |
| Air temperature – Supraglacial lake (°C/km$^2$) | 0.68 (5) | | 0.58 (7) | 0.66 (9) | 0.67 (4) | 0.79 (7) |
| Melt – Velocity at Urdukas (km$^2$d/m) | 0.18 (936) | | 0.53 (2946) | 0.36 (-1085) | 0.35 (5149) | 0.0 (246) |
| Melt – Velocity at Yermanendu (km$^2$d/m) | 0.44 (700) | | 0.72 (1358) | 0.05 (-348) | 0.66 (3100) | |
| Melt – Velocity at Gore (km$^2$d/m) | 0.78 (831) | 0.28 (839) | 0.70 (1164) | | 0.74 (1215) | |
| Melt – Velocity at Concordia (km$^2$d/m) | 0.87 (972) | 0.41 (588) | 0.80 (1059) | 0.16 (253) | 0.90 (1031) | |
| Supraglacial lake – Velocity at Urdukas (km$^2$d/m) | 0.0 (3) | 0.07 (1) | 0.25 (11) | 0.53 (-8) | | 0.14 (9) |
| Supraglacial lake – Velocity at Yermanendu (km$^2$d/m) | 0.18 (5) | 0.26 (3) | 0.67 (7) | 0.07 (-3) | 0.54 (27) | 0.30 (6) |
| Supraglacial lake – Velocity at Gore(km$^2$d/m) | 0.52 (7) | 0.53 (3) | 0.71 (6) | 0.0 (0) | 0.6 (12) | 0.30 (4) |
| Supraglacial lake – Velocity at Concordia (km$^2$d/m) | 0.91 (10) | 0.36 (1) | 0.94 (6) | | 0.75 (9) | 0.53 (4) |
| Precipitation – Velocity at Urdukas (mm d/m) | | | 0.31 (-255) | | 0.04 (-137) | |
| Precipitation – Velocity at Yermanendu (mm d/m) | 0.06 (-28) | 0.02 (49) | 0.27 (-96) | | 0.09 (-87) | |
| Precipitation – Velocity at Gore (mm d/m) | 0.08 (-28) | | 0.26 (-81) | | 0.06 (-28) | |
| Precipitation – Velocity at Concordia (mm d/m) | 0.07 (-30) | | 0.20 (-61) | | 0.07 (-22) | |
| Melt - Runoff-index (km$^2$/km$^2$) | 0.88 (367) | 0.28 (160) | 0.62 (465) | 0.60 (129) | 0.69 (277) | 0.62 (123) |
| Supraglacial lake - Runoff-index (km$^2$/km$^2$) | 0.77 (4) | | 0.62 (2) | 0.48 (1) | 0.59 (2) | 0.55 (1) |
| Velocity at Urdukas - Runoff-index (m/d km$^2$) | | 0.26 (0) | 0.46 (0) | 0.35 (0) | 0.35 (0) | |
| Velocity at Yermanendu - Runoff-index (m/d km$^2$) | 0.51 (0) | 0.04 (0) | 0.68 (0) | 0.22 (0) | 0.62 (0) | |
| Velocity at Gore - Runoff-index (m/d km$^2$) | 0.78 (0) | 0.0 (0) | 0.66 (0) | 0 (0.1) | 0.59 (0) | |
| Velocity at Concordia - Runoff-index (m/d km$^2$) | 0.88 (0) | 0.02 (0) | 0.67 (0) | | 0.72 (0) | |

## 5 Discussion

In this section, we discuss the relationship between glacier velocity, basal sliding and hydrological drainage as well as the impact of melt, supraglacial lakes and debris cover on this process.

### 5.1 Transition between efficient and inefficient subglacial drainage

Although each year exhibits a different behavior, we present in the following a general conceptual model that tries to explain the observed seasonal patterns. The transition of winter to spring was typically observed between the end of March and the end of April, as positive air temperatures begin to persist above 0°C for periods of multiple days up to 4700 m a.s.l (Fig. 2). The snow on the main branch usually has already melted to Gore at this time. This warm period in spring causes supraglacial lake formation up to 4300 m a.s.l. due to meltwater from snow and ice and the first snowmelt on the tributary glaciers and main

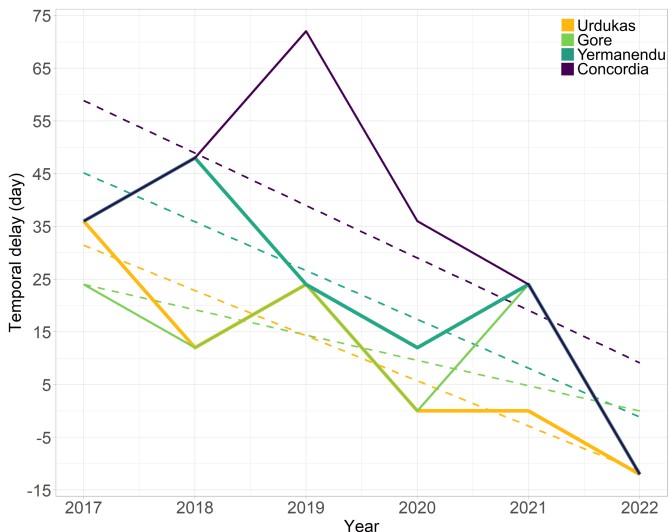

**Figure 5.** Temporal delay between first seasonal melt and glacier velocity peak for Urdukas, Yermanendu, Gore, and Concordia. The "linear trend" - to indicate the tendency only - is displayed in dotted lines.

branch up to 4700 m a.s.l. (Fig. 3 and Fig. 4a). The surface meltwater flows into the ice via crevasses and reaches the cavities at the glacier bed via vertical conduits. As the subglacial drainage system is inefficient, the channels cannot cope with the influx of the first melt causing a high subglacial water pressure and separation of ice from the bed with a subsequent basal motion (Mair et al., 2001; Macgregor et al., 2005; Burgess et al., 2013; Nanni et al., 2023). The high correlation between melt and velocity confirms this phenomenon (Tab. 2 and Tab. 3). The spring speed-up with a relative increase of 0.1-0.16 m d$^{-1}$ (30-35%) affects the complete glacier area between Urdukas and Concordia. In the years 2017, 2019, 2020, and 2021, the velocity at Gore exceeds that at Concordia suggesting that basal sliding is greatest at this location (Fig. 2). This phenomenon was already observable in 2008-2010 and 2014 (Wendleder et al., 2018).

In summer, persistently warm air temperatures support the formation and expansion of supraglacial lakes by producing large amounts of meltwater from snow and ice (Fig. 2). A detailed overview of the distribution and duration of the supraglacial lakes from 2017 to 2020 is shown in Wendleder et al. (2021a). With the rise in air temperatures, the melt expands up to 5600 m a.s.l. and provides a continuous water influx into the channels and continuous velocity increase to up to 0.59 m d$^{-1}$ from June to July. A notable indication of basal sliding during the summer months is the substantial increase, ranging from 2 to 6 times, in velocity compared to the winter (Mayer et al., 2006). These high summer values were also observed in summer 2004 and 2005 (Quincey et al., 2009) and summer 2008, 2009, and 2010 (Wendleder et al., 2018). The water pressure and the resulting frictional heat leading to a melting of the ice walls causes an expansion of the channels and thus, leads to a transition from an inefficient to an efficient subglacial drainage (Schoof, 2010; Sundal et al., 2011; Benn et al., 2017). The increase in the capacity of the drainage system reduces the water pressure and slows the glacier motion between Urdukas and Concordia down to the winter values of 0.25-0.39 m d$^{-1}$ (Fig. 2, Nienow et al., 1998; Hewitt, 2013; Vincent and Moreau, 2016).

Afterwards, an increase in the time delayed runoff at the glacier terminus is observable. Only early July 2022, an abrupt change from low to high runoff is recognizable indicating a faster transition to an efficient subglacial drainage (Fig. 2). In the other years, however, the runoff is linearly increasing or fluctuating gradually through time between minimum and maximum extent. Presumably, the higher summer velocities provokes the creation of crevasses and cracks and thus enables the drainage of the supraglacial lakes. The water of the lakes draining after the slowdown lead, in addition to the melt, to a water surplus and hence to an increase in water pressure and basal sliding during fall (Fig. 2, Tab. 2, and Tab. 3), but with lower magnitude by 0.1-0.2 m d$^{-1}$ (20-30%) than the summer velocity peak (Hewitt, 2013; Hart et al., 2022; Nanni et al., 2023).

## 5.2 Impact of melt and supraglacial lakes

In Wendleder et al. (2018), we inferred that warmer spring seasons (April-May) with higher precipitation or melt rates would lead to increased formation of supraglacial lakes and that the discharge of the supraglacial lakes would cause an increased basal sliding. Increased availability of Earth Observation data since 2016 has now enabled a continuous mapping of the seasonal evolution of supraglacial lakes (Wendleder et al., 2021a) and melt and thus provides an insight into the detailed process.

This study shows that snow and ice melt covers the largest area of 319-353 km$^2$ in mid-July to mid-August (Fig. 3). Assuming a snow depth of 1-1.5 m and the density for settled snow of 350 kg m$^{-3}$ at the end of the ablation season (Mayer et al., 2014), this could roughly estimated to a snow water equivalent of 122-188 million m$^3$. However, the supraglacial lakes cover a total annual area of 3.6-5.9 km$^2$ and a maximal volume of 60 million m$^3$ assuming an average water depth of 10 m (Liu et al., 2015). These estimates show that the water volume of the melt is twice as large as that of the lakes and therefore has a significantly greater influence on the velocity and to the transition to an efficient subglacial drainage.

## 5.3 Impact of debris cover

Debris-covered glaciers show a different response to global warming, because ice melt is reduced by the insulating effect of the debris cover over extensive part of the ablation area (in case the debris reaches a critical thickness). Another characteristic is their ability to store large amounts of water in supraglacial lakes, which are suddenly released when crevasses opens, often resulting in basal sliding.

The important findings of this study are 1) the period of several days with positive air temperatures in spring lead to an increase in lake area and to a melt up to 4700 m a.s.l., 2) the large melting area of snow and ice causing the spring speed-up and the high summer glacier velocities, and 3) the fact that the Baltoro glacier also shows an acceleration in fall, caused by the drained supraglacial lakes. The area affected by the melt and its effects on glacier dynamics are therefore surprisingly large. Despite the insulation due to the debris cover on the main branch of the Baltoro Glacier, the ice cliffs and supraglacial lakes act as hotspots that react sensitively to the rise in temperature and lead to an additional ice ablation.

# 6 Conclusions

To gain a more comprehensive understanding of whether, and how, meltwater and lake drainage events trigger or contribute to basal sliding, we conducted a spatio-temporal analysis. This involved integrating glaciological variables such as surface velocity, supraglacial lake extent, snow and ice melt, and runoff-index derived from Earth Observation data, along with climate variables including air temperature and precipitation obtained from the HAR dataset. The multi-parameter time series was analysed by the Pearson correlation and linear regression. Our study showed that the period of several days with positive air temperatures from the end of March to the end of April led to an abrupt transition from winter to spring and that the ablation period was influenced by continued, high air temperatures (Fig. 2). The higher air temperatures affected both snow and ice melt as well as supraglacial lake formation (Fig. 4a). The snow and ice melt of the tributary glaciers, covering up to 64 % of the total area of Baltoro Glacier (Fig. 3b) and lasting from April to November/December (Fig. 2), had the greatest impact to the basal sliding that led to the spring speed-up of 0.1-0.16 m d$^{-1}$ (30-35%) in relation to the winter velocities and the summer velocity increase by 0.1-0.2 m d$^{-1}$ (20-50%) in relation to the spring velocities (Tab. A1 and A2). The subsequent transition from inefficient to efficient subglacial drainage lead to a glacier slow down. Afterwards, the supraglacial lakes with a total area of 3.6-5.9 km$^2$ (Fig. 3b) contribute to the fall speed-up which has a lower magnitude by 0.1-0.2 m d$^{-1}$ (20-30%) than the summer velocity peak. The discharge from snow and ice melt accounts for the largest amount of runoff (Tab. A1 and A2).

The year 2022 exhibits notable climatic impacts in the form of a sustained warm period (Tab. 1), giving rise to a series of consequential phenomena. This includes an increased number of supraglacial lakes, an advancement in the timing of maximal lake area, and a substantial amplification of snow and ice melt (Tab. 1). These factors collectively provoke two separate peaks in surface velocities during spring and summer (Fig. 2f), driven predominantly by the mechanism of basal sliding. Additionally, this altered dynamic triggers a more rapid transition towards an efficient subglacial drainage system (Fig. 2f).

There exists an indication that the temporal delay between initial peak of seasonal snow and ice melt and first relative velocity maximum is decreasing. This observation warrants a careful interpretation, suggesting that the glacier's response is intensifying in relation to the processes of snow and ice melt and velocity. Consequently, this increases the glacier's potential for stronger responsiveness to climatic variations, in particular elevated air temperatures, thereby implying an augmented susceptibility to the effects of climate change.

*Code and data availability.* The code is available on request. The TerraSAR-X data are available through the DLR (©DLR 2019-2022), Sentinel-1 and 2 data are provided by the DLR Processing and Archiving Center (PAC) / Long-Term Archive (LTA) (©ESA 2019-2022), PlanetScope data are available through Planet (©Planet 2019-2022), and the High Asia Refined analysis (HAR) dataset by the Chair of Climatology, TU Berlin.

*Author contributions.* AW, AS, DJQ, CM, and MB designed the research. AW analyzed the data and results and wrote the manuscript. JB helped with the spatial representation, JI calculated the surface velocity fields, TE determined the temporal delay, and PD processed the MAJA corrected Sentinel-2 data. All authors helped to edit and to improve the manuscript.

*Competing interests.* The authors declare that they have no competing interests

*Acknowledgements.* We would like to thank both anonymous reviewers for their corrections that helped to strengthen the manuscript.

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

**Table A1.** Pearson correlation (r) of glacier and climate variables for 2017-2019. Only significant correlations according to a 95 % confidence interval are shown.

| | Period I 2017 21.03.-04.06. | Period II 2017 05.06.-30.09. | Period I 2018 12.03.-17.07. | Period II 2018 18.07.-30.09. | Period I 2019 27.03.-12.07. | Period II 2019 13.07.–30.09. |
|---|---|---|---|---|---|---|
| Air temperature – Melt | 0.70 | 0.81 | 0.83 | 0.84 | 0.80 | 0.83 |
| Air temperature – Supraglacial lake | 0.58 | 0.71 | 0.87 | 0.89 | 0.70 | 0.69 |
| Melt – Velocity at Urdukas | 0.94 | 0.30 | -0.30 | -0.24 | -0.21 | -0.19 |
| Melt – Velocity at Yermanendu | 0.92 | 0.36 | 0.08 | -0.37 | 0.09 | 0.58 |
| Melt – Velocity at Gore | 0.93 | 0.33 | 0.63 | -0.01 | 0.60 | 0.69 |
| Melt – Velocity at Concordia | 0.92 | 0.45 | 0.77 | 0.36 | 0.92 | 0.80 |
| Supraglacial lake – Velocity at Urdukas | 0.91 | -0.14 | -0.13 | -0.09 | 0.02 | -0.42 |
| Supraglacial lake – Velocity at Yermanendu | 0.86 | 0.48 | 0.07 | -0.28 | 0.33 | 0.37 |
| Supraglacial lake – Velocity at Gore | | 0.63 | 0.80 | 0.12 | 0.77 | 0.68 |
| Supraglacial lake – Velocity at Concordia | | 0.69 | 0.93 | 0.52 | 0.96 | 0.87 |
| Precipitation – Velocity at Urdukas | -0.46 | -0.05 | 0.15 | 0.14 | -0.02 | 0.21 |
| Precipitation – Velocity at Yermanendu | -0.47 | -0.53 | 0.08 | 0.22 | -0.04 | 0.12 |
| Precipitation – Velocity at Gore | -0.47 | 0.07 | -0.17 | 0.13 | -0.10 | 0.03 |
| Precipitation – Velocity at Concordia | -0.45 | 0.07 | -0.22 | -0.02 | -0.14 | -0.01 |
| Melt - Runoff-index | 0.80 | 0.65 | 0.88 | 0.93 | 0.85 | 0.75 |
| Supraglacial lake - Runoff-index | 0.72 | 0.63 | 0.94 | 0.90 | 0.80 | 0.51 |
| Velocity at Urdukas - Runoff-index | 0.85 | -0.12 | -0.02 | -0.35 | -0.03 | 0.38 |
| Velocity at Yermanendu - Runoff-index | 0.91 | -0.03 | 0.08 | -0.48 | 0.25 | 0.79 |
| Velocity at Gore - Runoff-index | 0.90 | -0.13 | 0.81 | -0.13 | 0.68 | 0.64 |
| Velocity at Concordia - Runoff-index | 0.85 | -0.04 | | 0.24 | 0.81 | 0.57 |

**Table A2.** Pearson correlation (r) of glacier and climate variables for 2020-2022. Only significant correlations according to a 95 % confidence interval are shown.

| | Period I 2020 21.03.-24.06. | Period II 2020 25.06.-30.09. | Period I 2021 07.03.-19.06. | Period II 2021 20.06.-30.09. | Period I 2022 05.03.-08.07. | Period II 2022 09.07.–30.09. |
|---|---|---|---|---|---|---|
| Air temperature – Melt | 0.82 | 0.91 | 0.78 | 0.71 | 0.89 | 0.90 |
| Air temperature – Supraglacial lake | 0.82 | 0.42 | 0.76 | 0.72 | 0.80 | 0.89 |
| Melt – Velocity at Urdukas | 0.44 | 0.22 | 0.73 | -0.60 | 0.59 | 0.11 |
| Melt – Velocity at Yermanendu | 0.68 | 0.28 | 0.85 | 0.26 | 0.81 | 0.15 |
| Melt – Velocity at Gore | 0.89 | 0.52 | 0.84 | -0.05 | 0.86 | 0.14 |
| Melt – Velocity at Concordia | 0.93 | 0.64 | 0.89 | 0.40 | 0.95 | 0.32 |
| Supraglacial lake – Velocity at Urdukas | 0.14 | 0.28 | 0.51 | -0.87 | 0.36 | 0.38 |
| Supraglacial lake – Velocity at Yermanendu | 0.44 | 0.52 | 0.82 | -0.51 | 0.74 | 0.55 |
| Supraglacial lake – Velocity at Gore | 0.73 | 0.72 | 0.85 | -0.38 | 0.83 | 0.56 |
| Supraglacial lake – Velocity at Concordia | 0.96 | 0.60 | 0.98 | 0.15 | 0.86 | 0.73 |
| Precipitation – Velocity at Urdukas | -0.19 | 0.17 | -0.56 | -0.03 | -0.22 | 0.03 |
| Precipitation – Velocity at Yermanendu | -0.26 | 0.18 | -0.53 | 0.02 | -0.32 | -0.06 |
| Precipitation – Velocity at Gore | -0.30 | 0.05 | -0.52 | 0.07 | -0.28 | -0.06 |
| Precipitation – Velocity at Concordia | -0.29 | -0.05 | -0.45 | 0.12 | -0.28 | -0.06 |
| Melt - Runoff-index | 0.93 | 0.53 | 0.79 | 0.77 | 0.84 | 0.82 |
| Supraglacial lake - Runoff-index | 0.88 | -0.11 | 0.78 | 0.75 | 0.75 | 0.85 |
| Velocity at Urdukas - Runoff-index | 0.50 | 0.51 | 0.68 | -0.59 | 0.59 | 0.25 |
| Velocity at Yermanendu - Runoff-index | | 0.22 | | -0.51 | 0.79 | 0.24 |
| Velocity at Gore - Runoff-index | 0.89 | -0.15 | 0.82 | -0.36 | 0.77 | 0.23 |
| Velocity at Concordia - Runoff-index | 0.94 | -0.21 | 0.82 | -0.03 | | 0.38 |

**Table A3.** Observation Period of glacier surface velocities derived from Sentinel-1 (orbit 27). Indicated are the date of the first and second acquisition used to derive the glacier velocity.

| Oct 2016 - Sept 2017 | Oct 2017 - Sept 2018 | Oct 2018 - Sept 2019 | Oct 2019 - Sept 2020 | Oct 2020 - Sept 2021 | Oct 2021 - Sept 2022 |
|---|---|---|---|---|---|
| 20.10.2016 - 13.11.2016 | 21.09.2017 - 15.10.2017 | 28.09.2018 - 22.10.2018 | 23.09.2019 - 17.10.2019 | 29.09.2020 - 23.10.2020 | 24.09.2021 - 18.10.2021 |
| 13.11.2016 - 07.12.2016 | 03.10.2017 - 27.10.2017 | 10.10.2018 - 03.11.2018 | 05.10.2019 - 29.10.2019 | 11.10.2020 - 04.11.2020 | 06.10.2021 - 30.10.2021 |
| 07.12.2016 - 31.12.2016 | 15.10.2017 - 08.11.2017 | 22.10.2018 - 15.11.2018 | 17.10.2019 - 10.11.2019 | 23.10.2020 - 16.11.2020 | 18.10.2021 - 11.11.2021 |
| 31.12.2016 - 24.01.2017 | 27.10.2017 - 20.11.2017 | 03.11.2018 - 27.11.2018 | 29.10.2019 - 22.11.2019 | 04.11.2020 - 28.11.2020 | 30.10.2021 - 23.11.2021 |
| 24.01.2017 - 17.02.2017 | 08.11.2017 - 02.12.2017 | 15.11.2018 - 09.12.2018 | 10.11.2019 - 04.12.2019 | 16.11.2020 - 10.12.2020 | 11.11.2021 - 05.12.2021 |
| 05.02.2017 - 01.03.2017 | 20.11.2017 - 14.12.2017 | 27.11.2018 - 21.12.2018 | 22.11.2019 - 16.12.2019 | 28.11.2020 - 22.12.2020 | 23.11.2021 - 17.12.2021 |
| 17.02.2017 - 13.03.2017 | 02.12.2017 - 26.12.2017 | 09.12.2018 - 02.01.2019 | 04.12.2019 - 28.12.2019 | 10.12.2020 - 03.01.2021 | 05.12.2021 - 29.12.2021 |
| 01.03.2017 - 25.03.2017 | 14.12.2017 - 07.01.2018 | 21.12.2018 - 14.01.2019 | 16.12.2019 - 09.01.2020 | 22.12.2020 - 15.01.2021 | 29.12.2021 - 22.01.2022 |
| 13.03.2017 - 06.04.2017 | 26.12.2017 - 19.01.2018 | 02.01.2019 - 26.01.2019 | 28.12.2019 - 21.01.2020 | 03.01.2021 - 27.01.2021 | 22.01.2022 - 15.02.2022 |
| 25.03.2017 - 18.04.2017 | 19.01.2018 - 12.02.2018 | 14.01.2019 - 07.02.2019 | 09.01.2020 - 02.02.2020 | 15.01.2021 - 08.02.2021 | 03.02.2022 - 27.02.2022 |
| 06.04.2017 - 30.04.2017 | 12.02.2018 - 08.03.2018 | 14.01.2019 - 07.02.2019 | 21.01.2020 - 14.02.2020 | 27.01.2021 - 20.02.2021 | 15.02.2022 - 11.03.2022 |
| 18.04.2017 - 12.05.2017 | 24.02.2018 - 20.03.2018 | 26.01.2019 - 19.02.2019 | 02.02.2020 - 26.02.2020 | 08.02.2021 - 04.03.2021 | 27.02.2022 - 23.03.2022 |
| 30.04.2017 - 24.05.2017 | 08.03.2018 - 01.04.2018 | 07.02.2019 - 03.03.2019 | 14.02.2020 - 09.03.2020 | 20.02.2021 - 16.03.2021 | 11.03.2022 - 04.04.2022 |
| 12.05.2017 - 05.06.2017 | 20.03.2018 - 13.04.2018 | 19.02.2019 - 15.03.2019 | 26.02.2020 - 21.03.2020 | 04.03.2021 - 28.03.2021 | 23.03.2022 - 16.04.2022 |
| 05.06.2017 - 29.06.2017 | 13.04.2018 - 07.05.2018 | 03.03.2019 - 27.03.2019 | 09.03.2020 - 02.04.2020 | 16.03.2021 - 09.04.2021 | 04.04.2022 - 28.04.2022 |
| 29.06.2017 - 23.07.2017 | 07.05.2018 - 31.05.2018 | 15.03.2019 - 08.04.2019 | 21.03.2020 - 14.04.2020 | 28.03.2021 - 21.04.2021 | 16.04.2022 - 10.05.2022 |
| 11.07.2017 - 04.08.2017 | 19.05.2018 - 12.06.2018 | 27.03.2019 - 20.04.2019 | 02.04.2020 - 26.04.2020 | 09.04.2021 - 03.05.2021 | 28.04.2022 - 22.05.2022 |
| 23.07.2017 - 16.08.2017 | 31.05.2018 - 24.06.2018 | 08.04.2019 - 02.05.2019 | 14.04.2020 - 08.05.2020 | 21.04.2021 - 15.05.2021 | 10.05.2022 - 03.06.2022 |
| 04.08.2017 - 28.08.2017 | 24.06.2018 - 18.07.2018 | 20.04.2019 - 14.05.2019 | 26.04.2020 - 20.05.2020 | 03.05.2021 - 27.05.2021 | 22.05.2022 - 15.06.2022 |
| 16.08.2017 - 09.09.2017 | 18.07.2018 - 11.08.2018 | 02.05.2019 - 26.05.2019 | 08.05.2020 - 01.06.2020 | 15.05.2021 - 08.06.2021 | 03.06.2022 - 27.06.2022 |
| 28.08.2017 - 21.09.2017 | 30.07.2018 - 23.08.2018 | 14.05.2019 - 07.06.2019 | 20.05.2020 - 13.06.2020 | 27.05.2021 - 20.06.2021 | 15.06.2022 - 09.07.2022 |
| 09.09.2017 - 03.10.2017 | 11.08.2018 - 04.09.2018 | 26.05.2019 - 19.06.2019 | 01.06.2020 - 25.06.2020 | 08.06.2021 - 02.07.2021 | 27.06.2022 - 21.07.2022 |
|  | 23.08.2018 - 16.09.2018 | 07.06.2019 - 01.07.2019 | 13.06.2020 - 07.07.2020 | 20.06.2021 - 14.07.2021 | 09.07.2022 - 02.08.2022 |
|  | 04.09.2018 - 28.09.2018 | 19.06.2019 - 13.07.2019 | 25.06.2020 - 19.07.2020 | 02.07.2021 - 26.07.2021 | 21.07.2022 - 14.08.2022 |
|  | 16.09.2018 - 10.10.2018 | 01.07.2019 - 25.07.2019 | 07.07.2020 - 31.07.2020 | 14.07.2021 - 07.08.2021 | 02.08.2022 - 26.08.2022 |
|  |  | 13.07.2019 - 06.08.2019 | 19.07.2020 - 12.08.2020 | 26.07.2021 - 19.08.2021 | 14.08.2022 - 07.09.2022 |
|  |  | 25.07.2019 - 18.08.2019 | 31.07.2020 - 24.08.2020 | 07.08.2021 - 31.08.2021 | 26.08.2022 - 19.09.2022 |
|  |  | 06.08.2019 - 30.08.2019 | 24.08.2020 - 17.09.2020 | 19.08.2021 - 12.09.2021 | 07.09.2022 - 01.10.2022 |
|  |  | 18.08.2019 - 11.09.2019 | 05.09.2020 - 29.09.2020 | 31.08.2021 - 24.09.2021 |  |
|  |  | 30.08.2019 - 23.09.2019 | 17.09.2020 - 11.10.2020 |  |  |
|  |  | 11.09.2019 - 05.10.2019 |  |  |  |