# Peer review of "Velocity variations and hydrological drainage at Baltoro Glacier, Pakistan"

_The Cryosphere, 2023_

## Referee Comment (RC1)

**Review of Wendleder et al.: Basal Sliding and Hydrological Drainage at Baltoro Glacier**

Wendleder et al. investigate the climate-hydrology interactions of the debris-covered Baltoro Glacier using a broad suite of remote sensing data products to further understand the debris-covered glacier cycle generally and gain insight into this glacier system more specifically. The authors combine meteorological (air temperature, precipitation) and glaciological (surface velocity, supraglacial lake area, snowmelt area, and proglacial runoff) data for six complete hydrological years (2016-2022). This wide variety of data is used to interpret hydrological changes responsible for the observed surface velocity patterns. Snowmelt and supraglacial lake drainage are found to be the most influential factors for explaining surface velocity patterns. The patterns themselves significantly vary year-to-year, with the warmest year (2022) displaying behaviour that is distinct from the other cooler years.

The authors have done a good job at integrating an impressive breadth of high temporal resolution (and some spatially resolved) data to answer their primary questions and have clearly presented the data in a few very useful figures. My main comment is that, while the data used in the study have been appropriately summarized and presented, the results and discussion do not synthesize and summarize the data to the same level. I feel that the manuscript will be a valuable contribution to the debris-covered glacier literature, as well as the broader literature investigating melt-hydrology-dynamics feedbacks more generally, after some work to more effectively summarize the results and extract interpretable patterns.

I have listed these and other general comments below along with a number of specific comments. The authors should interpret the number of specific comments as an indicator of my interest in the work. I hope that the authors find most of the comments are reasonable, and I understand that not all the open-ended comments may be addressed.

**General Comments**

1. Clarity and conciseness in Results. I feel that with some work the Results can be brought up to the standard set by the comprehensive data set developed in the methods section. In the current form, there is a large quantity of information presented with little structure to help with understanding. My general comment would be to develop a concise list of the few most important points from each type of analysis and to focus the results and discussion on these.

   I would also encourage the authors to further divide each large section (e.g., 4.1 Temporal Relationship) into sub-sub-sections, with one sub-sub-section for each data product/relationship. I think this would help readers find relevant information they are looking for. Each of these sections could further be structured to begin with the most broad patterns, and progressively focus down towards the most specific.

2.  I think the scope of the work could be expanded by including a stronger link to the impact of debris cover on the inferred hydrology and velocity patterns, and how this may be transferable to other debris-covered systems. This goal is hinted at in the introduction by pointing out some differences between debris-free and debris-covered glacier behaviour, and it would be nice to see this continued through the manuscript.

3.  I have some questions about the conclusion that the time lag between snowmelt and velocity is decreasing, and whether this is fully supported by the data and results. I'm not clear on what results in particular support this statement. I would have thought this time lag would be quite sensitive to the particular pattern of surface melt each year, and so it would be difficult to conclude there is a persistent, large-scale climate-driven cause given the six years of data presented here. If you can support this statement with your results, it would be important to discuss some mechanisms that could explain such a trend in the Discussion, and how interannual snowmelt variability might make this challenging.

**Specific Comments**

May be nice to highlight in the abstract which of the identified behaviours are different due to debris cover. Perhaps the lakes?

Line 140-157: Do the authors have any assessment of the snowmelt mapping method, from this or other studies? It could be helpful to comment on the performance of the method over a variety of surface types, since the snowmelt data form a core part of the results and conclusions of this paper.

Line 165: You explain that you use a weather station located at Urdukas to derive a monthly mean lapse rate. Could you elaborate on the instrumentation and methods used for this? Do you have multiple temperature sensors at different elevations to derive the lapse rate, or are you comparing the point AWS measurement to the reanalysis data? Finally, can you comment on the quality of the HAR precipitation and temperature data compared based on the AWS data?

Figure 2 and related text: I generally find this figure an excellent visualization for the large quantity of data that it presents. I have two questions about the relationship between lake surface area and velocity.
  ● Do the lakes mostly drain into the subglacial drainage system through moulins and crevasses, so that we should interpret changes in lake area directly as a proxy for inputs into the subglacial system, or does much of the meltwater drain laterally without accessing the bed?
  ● There is clearly a temporal relationship between decreasing lake area and the timing of the fall velocity peak (especially in 2017, 2018, and 2021). In 2022, it looks like lake area is consistently decreasing from 1 June until 30 September, while velocity has complex patterns. For example at Gore, velocity increases until about July, decreases until about

September, then increases until the end of September. What factors might be influencing this complex behaviour, especially considering the consistent high temperatures you show in 2022? And how do you think this impacts your linear statistics?

Figure 3: This has the potential to be a great figure. I find it difficult to see the difference in the shade of blue between (a) and (b) to interpret the difference in glacier velocity. It is also difficult to see the supraglacial lake boundaries, at first the lake outlines look like noise in the velocity field since they are plotted in the same colour. Maybe it would be easier to interpret if the velocity field and lakes were shown in different colours.
- I am also curious why it looks like the snowmelt begins in the mid-glacier, in the Concordia zone. Has all the snow melted below this point already by 13 April, but then why would we see snowmelt on the glacier tongue starting between 7 May and 18 July?
- What is the detectability of wet snow over debris, where perhaps the snowpack can drain better through the debris layer, compared to bare ice where the snowpack may become saturated or swampy?

Figure 4: This is another great figure to display a lot of information. I think it might be nice to include a broader discussion of the patterns that you can see here (i.e., the difference in the shape between (a) and (b)) in the results section corresponding to this figure. For example, I see that there are generally lower velocity–snowmelt and velocity–runoff $R^2$ values in the second time period, but I found the results instead focussed on many specific years as examples. Even more broadly, most correlations are "shrunken" in (b), i.e. closer to the $R^2 = 0$ contour, showing that internal dynamics are more important later in the melt season.

Despite the title including "...and hydrological drainage", the hydrology piece is mostly not introduced until the Discussion. Perhaps there could be some more that you can introduce in the results to emphasize this piece of the study.

I think the content that is in the Discussion is mostly good, however I feel readers may be curious about some other features that you could add:
- What are some drivers of variability in the relationships that you observe? For example, there are quite large year-to-year changes in the $R^2$ values between velocity and runoff in Fig. 4(b), and statistically significant trends that switch sign year-to-year in Table 2. Is this just a response to the specific temperature, precipitation, etc. patterns, or is there more going on?
- Revisit the trends from the results and highlight what is (un)expected, novel, and specific to debris-covered glaciers. How do the patterns seen here depend on the debris cover?

Section 5.1 goes through an overview of the inferred hydrology of Baltoro Glacier. The presented distributed/channelized switch conceptual model is well understood, but I wonder if in this case there might be more going on. For example, might changes in bed connectivity (e.g., Andrews et al., 2014; Hoffman et al., 2016) or a more nuanced understanding of basal hydrology and sliding (e.g., Gilbert et al., 2022) also explain some of the inferred patterns? What

is the basal environment (hard bed, soft bed?), and might storage/drainage through till be important here? A similar comment could be made about the description near line 30.

I think it would help to add one or two sentences about how you infer sliding from surface velocities (perhaps in Section 5.1, to support some of the conclusions). I imagine you are assuming the deformational component does not change significantly through the year, and so the only remaining mechanism to explain the seasonal velocity variations is a change in basal velocity. Making this explicit would support the later conclusions nicely.

What controls the timing of lake drainage? Do lakes reach a threshold size and then drain, or is it more complex? See also line 327 comment.

Line 324: I am curious if it's possible to use the difference between snowmelt and the runoff index to try to quantify, or at least observe, a release of stored water when the drainage system connects and channels can evacuate the system. This is hinted at here, but it would be interesting to look into this in a little more detail.

Line 327: The idea that high summer velocities enable crevasse development and lake drainage has some nuance and subtlety (e.g., Poinar & Andrews, 2021). Have you looked at this hypothesis by estimating strain rates (as best as possible with the limited resolution from remote sensing data) to see if this is at least reasonable? Further, wouldn't this mechanism predict lakes to drain mostly during acceleration, whereas Figure 2 shows lake area sometimes decreasing while velocities also decrease? I do think this is a reasonable mechanism to propose, but some more care should be taken here.

Line 368: It would be interesting to see the difference between peak surface velocities and net surface displacement over the melt season and over the calendar year (especially as the authors have elsewhere cited Sundal et al., 2011). This would be a nice addition to the discussion, then could be referenced here.

**Technical corrections**

Line 28: Does Baltoro Glacier have basal motion in winter? I would tend to think of basal motion as a continuum, and not that basal sliding initiates from zero over winter, but rather that basal motion accelerates as effective pressure decreases. For example, this helps to explain why there are winter variations in velocity, otherwise the deformational component would have to be changing rapidly.

Line 30: I do acknowledge that there are different ways to explain subglacial drainage evolution. However, to some readers (especially modellers), "inefficient channels" is an unusual phrase. If the authors agree, consider changing to describe these channels as small, incipient, etc.

Line 33-35: "In the absence of meltwater the *ice- overburden* pressure is larger and ...". Would this be more clear to say that the *effective pressure* is larger (because water pressure decreases, not because ice overburden changes)?

Line 35: Is there evidence that regelation contributes to channel closure?

Line 35-39: I prefer to think that glaciers respond to the balance between driving and resistive stresses. Driving stress should vary slowly, since this is controlled by geometry. The resistive stress (especially basal traction in response to effective-pressure variations) instead can vary over short timescales.

Line 37: I wonder if maybe the 50% contribution of sliding to total glacier surface velocity has been taken out of context. For Greenland outlet glaciers, or surge type glaciers, basal motion can contribute well over 50% to velocity.

Line 48: Would it also be conceptually possible for supraglacial lake discharge to contribute to enhanced subglacial channel development, eventually reducing ice surface velocities?

Line 49: This is a long (but nicely detailed) paragraph. It might be easier for the reader to split into two paragraphs, separating background theory from details specific to Baltoro Glacier.

Line 74: What proportion of the debris covered area is enhancing vs. inhibiting surface melt?

Figure 1: Adding elevation labels, at least for the boundary between regions, and perhaps distance markers (as long as the figure does not become too busy) would help readers who are not familiar with this glacier.

Section 3.2: I understand that this method is detailed in a previous paper, and the description here is in appropriate detail. Can you very briefly comment on the sensitivity/performance of the different sensors used here?

Section 3.3: The term 'runoff-index' used on line 123 is a great way to acknowledge the simplified metric used here (as compared to runoff in units $m^3$ $s^{-1}$). I was confused about the type of runoff being measured until this section. Could identify that you are measuring *proglacial* runoff (i.e., as opposed to runoff through supraglacial streams) and briefly acknowledge the area index the first time you say 'runoff' (including abstract)?

Section 3.5: Since I am not familiar with the HAR data specifically, I found the hierarchy between the different levels of meteorological data confusing here.

Line 179: What proportion of lakes did not drain or fill during the observation period?

Line 185: Is the end of the ablation season defined as 30 September based on the water year starting on 1 October, or was 30 September determined to be the usual end of summer melt?

Section 3.8: Do you see lakes drain fully within the 2–4 day observation frequency? If so, does the interpolation artificially smooth this lake area signal, i.e. turn a discrete drainage event that happened at some time in the 2-4 day window into smooth linear decrease at daily resolution?

Table 1: I appreciate the level of detail in this table. What is the difference between "aggregate" and "cumulative" lake area? And are these areas presented just for active lakes?

Line 211: What factors do you think cause the hardly discernible time delay from lower to upper sectors here?

Figure 2 is an excellent comparison of so many fields of data. Is there a way to also label each year with its total positive degree days without making the figure too busy?

L321: Should this be "Schoof, 2010"?

L328: "The drained lake which is only disposable after the slowdown" is unclear.

**References**

Andrews, L. C., Catania, G. A., Hoffman, M. J., Gulley, J. D., Lüthi, M. P., Ryser, C., ... & Neumann, T. A. (2014). Direct observations of evolving subglacial drainage beneath the Greenland Ice Sheet. *Nature*, *514*(7520), 80-83.

Gilbert, A., Gimbert, F., Thøgersen, K., Schuler, T. V., & Kääb, A. (2022). A Consistent Framework for Coupling Basal Friction With Subglacial Hydrology on Hard-Bedded Glaciers. *Geophysical Research Letters*, *49*(13), e2021GL097507.

Hoffman, M. J., Andrews, L. C., Price, S. F., Catania, G. A., Neumann, T. A., Lüthi, M. P., ... & Morriss, B. (2016). Greenland subglacial drainage evolution regulated by weakly connected regions of the bed. *Nature communications*, *7*(1), 13903.

Poinar, K., & Andrews, L. C. (2021). Challenges in predicting Greenland supraglacial lake drainages at the regional scale. *The Cryosphere*, *15*(3), 1455-1483.

---

## Author Response (AR1)

**Review 1:**

Wendleder et al. investigate the climate-hydrology interactions of the debris-covered Baltoro Glacier using a broad suite of remote sensing data products to further understand the debris-covered glacier cycle generally and gain insight into this glacier system more specifically. The authors combine meteorological (air temperature, precipitation) and glaciological (surface velocity, supraglacial lake area, snowmelt area, and proglacial runoff) data for six complete hydrological years (2016-2022). This wide variety of data is used to interpret hydrological changes responsible for the observed surface velocity patterns. Snowmelt and supraglacial lake drainage are found to be the most influential factors for explaining surface velocity patterns. The patterns themselves significantly vary year-to-year, with the warmest year (2022) displaying behaviour that is distinct from the other cooler years.

The authors have done a good job at integrating an impressive breadth of high temporal resolution (and some spatially resolved) data to answer their primary questions and have clearly presented the data in a few very useful figures. My main comment is that, while the data used in the study have been appropriately summarized and presented, the results and discussion do not synthesize and summarize the data to the same level. I feel that the manuscript will be a valuable contribution to the debris-covered glacier literature, as well as the broader literature investigating melt-hydrology-dynamics feedbacks more generally, after some work to more effectively summarize the results and extract interpretable patterns.

I have listed these and other general comments below along with a number of specific comments. The authors should interpret the number of specific comments as an indicator of my interest in the work. I hope that the authors find most of the comments are reasonable, and I understand that not all the open-ended comments may be addressed.

Dear Reviewer,

Thank you very much for your careful review and the many constructive and solution-oriented comments. We really appreciate this:) and it helped a lot to improve the manuscript. We have considered and thoroughly implemented all of your suggestions, except for two for the following reasons: the calculation of the strain stress, which would be very interesting, but unfortunately out of scope of this work, and the inclusion of the positive degree days which would make the plot too confusing as a third x-axis would be needed due to the different value ranges.

Your reviews have been answered line by line. For a better readability of our response, the answers to the reviews and the corrections in the manuscript are shown in blue.

Kind regards,

Anna Wendleder with co-authors

**General Comments**

1. Clarity and conciseness in Results. I feel that with some work the Results can be brought up to the standard set by the comprehensive data set developed in the methods section. In the current form, there is a large quantity of information presented with little structure to help with understanding. My general comment would be to develop a concise list of the few most important points from each type of analysis and to focus the results and discussion on these.

I would also encourage the authors to further divide each large section (e.g., 4.1 Temporal Relationship) into sub-sub-sections, with one sub-sub-section for each data product/relationship. I think this would help readers find relevant information they are looking for. Each of these sections could further be structured to begin with the most broad patterns, and progressively focus down towards the most specific.

Thank you for your good and constructive ideas! We reconstructed the section "4.1. Temporal relationship" and divided it into new sub-sub-sections to focus more on the relationships and therefore on specific seasons:

> 4.1.1 Warm spring season
>
> 4.1.2 Spring and summer speed
>
> 4.1.3 Fall speed-up
>
> 4.1.4 Run-off

2. I think the scope of the work could be expanded by including a stronger link to the impact of debris cover on the inferred hydrology and velocity patterns, and how this may be transferable to other debris-covered systems. This goal is hinted at in the introduction by pointing out some differences between debris-free and debris-covered glacier behaviour, and it would be nice to see this continued through the manuscript.

You are right, a stronger link to debris cover would enhance the manuscript and emphasize the characteristic of debris covered glaciers. We inserted additional explanations in the result:

"The debris cover leads to heterogeneous ablation on the glacier surface with enhanced melting during warm periods on the ice cliffs and the supraglacial lakes." (Line 228)

"Supraglacial lakes store large quantities of water and can hence influence the velocity pattern in the event of rapid drainage. The maximum area of the supraglacial lakes ranged between 1.9 km$^2$ (2019) and 3.1 km$^2$ (2022). The drainage started early-June to mid-July." (Line 264)

And in the discussion, we added the section "Impact of debris cover":

"Debris covered glaciers show a different response to global warming, because ice melt is reduced by the insulating effect of the debris cover over extensive part of the ablation area (in case the debris reaches a critical thickness). Another characteristic of debris-covered glacier is their ability to store large amounts of water in supraglacial lakes, which are suddenly released when crevasses open, often resulting in basal sliding.

The important findings of this study are 1) the period of several days with positive air temperatures in spring lead to an increase in lake rise and to a snowmelt up to 4700 m a.s.l., 2) the large melting area of snow and ice causing the spring speed-up and the high summer glacier velocities, and 3) the fact that the Baltoro glacier also shows an acceleration in fall, caused by the drained supraglacial lakes. The area affected by the melt and its effects on glacier dynamics is therefore surprisingly large. Despite the insulation due to the debris cover on the main branch of the Baltoro Glacier, the ice cliffs and supraglacial lakes act as hotspot that react sensitively to the rise in temperature and lead to an additional ice ablation." (Line 374)

3. I have some questions about the conclusion that the time lag between snowmelt and velocity is decreasing, and whether this is fully supported by the data and results. I'm not clear on what results

in particular support this statement. I would have thought this time lag would be quite sensitive to the particular pattern of surface melt each year, and so it would be difficult to conclude there is a persistent, large-scale climate-driven cause given the six years of data presented here. If you can support this statement with your results, it would be important to discuss some mechanisms that could explain such a trend in the Discussion, and how interannual snowmelt variability might make this challenging.

We agree that this aspect was not presented clear enough in our manuscript. Therefore, to support the conclusion that the time lag between the first seasonal snowmelt and the velocity peak is indicatively decreasing we have added Fig. 5. Accordingly, we are convinced that the time series plot, showing the temporal development of the time lag from 2017 to 2022 and its tendency of decrease, will better support the interpretation of the data and the conclusions drawn. In addition, we have strengthened the discussion of possible mechanisms in the manuscript as requested.

**Specific Comments**

May be nice to highlight in the abstract which of the identified behaviours are different due to debris cover. Perhaps the lakes?

Please accept our apologies on this, but we had to shorten the abstract in response to Reviewer 2's suggestions, and the explanation of the debris-covered glacier had to be removed. However, we will go into more detail in the introduction (Line 42)

Line 140-157: Do the authors have any assessment of the snowmelt mapping method, from this or other studies? It could be helpful to comment on the performance of the method over a variety of surface types, since the snowmelt data form a core part of the results and conclusions of this paper.

Sure! In Line 162, we added the assessment results of Nagler et al. (2000): "This approach achieves an overall accuracy of 81.4%-85.4% (Nagler and Rott, 2000)."

Line 165: You explain that you use a weather station located at Urdukas to derive a monthly mean lapse rate. Could you elaborate on the instrumentation and methods used for this? Do you have multiple temperature sensors at different elevations to derive the lapse rate, or are you comparing the point AWS measurement to the reanalysis data? Finally, can you comment on the quality of the HAR precipitation and temperature data compared based on the AWS data?

The AWS was equipped with only one temperature sensor (thermo-hygrometer with radiation shield) at a height of 2 m. It measured the temperature every hour (Mihalcea et al., 2006).

For the downscaling, we used the temperature measured at 12 noon, calculated the monthly mean value on basis of a fitting curve (3rd order) and compared it with the mean monthly HAR value at 12 noon (fitting curve with 3rd order). The result is a lapse rate for each month. The comparison of the HAR temperature data with in-situ measurements achieved a mean bias of -0.58 K ($r^2$=0.99, Wang et al., 2020).

This information has been added to the text accordingly (Line 174): "The air temperature data (2 m height) from an automatic weather station located at Urdukas (Mihalcea et al., 2008) needed for the local downscaling of the HAR data. The weather station is equipped with an air temperature sensor (thermo-hygrometer with radiation shield) taking hourly measurements for 2011. … Compared with in-situ data, the modelled air temperatures show a mean bias of -0.58 K (Wang et al., 2021). … To downscale the air temperatures, a mean lapse rate for each month was determined based on the 12

pm (noon) air temperatures data from the automatic weather station using a fitting curve (3rd order). The derived lapse rates varied between 7° C km-1 (December) and 11° C km-1 (June).”

Figure 2 and related text: I generally find this figure an excellent visualization for the large quantity of data that it presents. I have two questions about the relationship between lake surface area and velocity.

● Do the lakes mostly drain into the subglacial drainage system through moulins and crevasses, so that we should interpret changes in lake area directly as a proxy for inputs into the subglacial system, or does much of the meltwater drain laterally without accessing the bed?

Thank you so much for this hint! We added the following explanation to clarify this aspect (Line 243): “In the area upwards of Urdukas which is characterized by less debris cover, the meltwater accumulates in the supraglacial lakes and drains into the subglacial system or flows laterally into larger streams. In the area downstream of Urdukas, there are no larger supraglacial channels and the water can drain through moulins and crevasses to the ground.”

● There is clearly a temporal relationship between decreasing lake area and the timing of the fall velocity peak (especially in 2017, 2018, and 2021). In 2022, it looks like lake area is consistently decreasing from 1 June until 30 September, while velocity has complex patterns. For example, at Gore, velocity increases until about July, decreases until about September, then increases until the end of September. What factors might be influencing this complex behaviour, especially considering the consistent high temperatures you show in 2022? And how do you think this impacts your linear statistics?

We assume that the prolonged heat wave in spring 2022 (Otto et al., 2023) led to an increased release of meltwater which could result in an earlier transition from an inefficient to an efficient drainage system and thus a slow-down of the glacier from July onwards. By means of the Earth observation data, we are only able to determine the area of snowmelt and supraglacial lakes, but not how much snow water equivalent was released or the lake volume compared to previous years. Therefore, the statistics cannot reflect this phenomenon either.

We added the assumption of an earlier transition to the results (Line 278): “…2022 by a continuous low runoff-index until the end of June and a sudden rise and thus to a distinct transition to an efficient drainage.”

Figure 3: This has the potential to be a great figure. I find it difficult to see the difference in the shade of blue between (a) and (b) to interpret the difference in glacier velocity. It is also difficult to see the supraglacial lake boundaries, at first the lake outlines look like noise in the velocity field since they are plotted in the same colour. Maybe it would be easier to interpret if the velocity field and lakes were shown in different colours.

Figure 3 has been renewed accordingly. It now includes the locations of Urdukas, Yermanendu, Gore, and Concordia, the colours of the glacier velocity has been changed to better discriminate them from the supraglacial lakes and the outline of the supraglacial lakes has been intensified.

[Figure]

● I am also curious why it looks like the snowmelt begins in the mid-glacier, in the Concordia zone. Has all the snow melted below this point already by 13 April, but then why would we see snowmelt on the glacier tongue starting between 7 May and 18 July?

Thanks for picking this up - we checked the snowmelt results and realized that there was a classification error in the snowmelt result from 18 July. There is no snowmelt over the glacier tongue between 7 May and 18 July. The error was corrected and Figure 3 was renewed.

● What is the detectability of wet snow over debris, where perhaps the snowpack can drain better through the debris layer, compared to bare ice where the snowpack may become saturated or swampy?

Microwaves strongly respond to changes in the liquid water content. At air temperatures of 0° C, snow can support water in liquid form which leads to an absorption of the SAR signal at the upper snow surface. Since the liquid water content in the upper snow surface is relevant, it has no influence on whether the meltwater can drain better over a debris layer than over bare ice.

In addition to the water content, the roughness has an influence on the SAR signal, too. A greater surface roughness leads to a higher SAR backscatter (dihedral surface scattering), if the surface roughness is greater than the wavelength (in our case 5.6 cm) and if the surface objects are oriented to the SAR sensor. Hence, wet snow over debris could have a higher SAR backscatter than wet snow over bare ice. However, this phenomenon was not observed in this study.

Figure 4: This is another great figure to display a lot of information. I think it might be nice to include a broader discussion of the patterns that you can see here (i.e., the difference in the shape between (a)

and (b)) in the results section corresponding to this figure. For example, I see that there are generally lower velocity–snowmelt and velocity–runoff R 2 values in the second time period, but I found the results instead focussed on many specific years as examples. Even more broadly, most correlations are "shrunken" in (b), i.e. closer to the R 2 = 0 contour, showing that internal dynamics are more important later in the melt season.

Thank you for your great idea! We already discussed how to present this section better, but to compare the two periods makes absolutely sense. We have completely rewritten the section focusing more on the differences. We hope that this part is now easier to read and to follow (Line 295):

"In the two periods, the correlation of the variables temperature and melt (0.7-0.95) as well as temperature and supraglacial lake (0.58-0.89) showed a similar pattern with a high relationship (>0.6) though with negligible minor changes in the Pearson correlation during both periods (0.1-0.2); except for the decrease by 0.4 for temperatures and supraglacial lakes in the second period of 2020. A high degree of correlation existed between melt and runoff-index (0.65–0.93) and supraglacial lakes (0.63–0.94), respectively, albeit with outliers in both relationships in the second period of 2020 (decrease by 0.4 and 0.9). The relationship between supraglacial lakes and velocity as well as melt and velocity showed the same pattern with higher correlations at Concordia and Yermanendu and lower correlations in Urdukas in both periods. However, the correlations of the second period vary significantly (0.5 to -0.8) over the years and generally decrease by 0.4-1.3 in the second period. Particularly noticeable was the large decline (<1.5) in Urdukas in both parameters. The correlation between velocity and runoff decreased by 1.3 in the second period. However, compared to the two pairs of variables mentioned above, the second period is characterised by a heterogeneous pattern over the years."

Despite the title including "...and hydrological drainage", the hydrology piece is mostly not introduced until the Discussion. Perhaps there could be some more that you can introduce in the results to emphasize this piece of the study.

Thank you for the suggestions. In the results, we referred to the hydrology as follows:

"In the area upwards of Urdukas, which is characterized by less debris cover, the meltwater accumulates in the supraglacial lakes and drains into the subglacial system or flows laterally into larger streams. In the area downstream of Urdukas, there are no larger supraglacial channels and the water can drain through moulins and crevasses to the ground." (Line 243)

"The runoff intensity is an indicator of when the water began to flow continuously and thus marks the transition from inefficient to efficient drainage…2022 by a continuous low runoff-index until the end of June and a sudden rise and thus to a distinct transition to an efficient drainage." (Line 271)

I think the content that is in the Discussion is mostly good, however I feel readers may be curious about some other features that you could add:

● What are some drivers of variability in the relationships that you observe? For example, there are quite large year-to-year changes in the R 2 values between velocity and runoff in Fig. 4(b), and statistically significant trends that switch sign year-to-year in Table 2. Is this just a response to the specific temperature, precipitation, etc. patterns, or is there more going on?

Thank you very much for spotting this! We assume that air temperature, precipitation, and solar radiation are higher than the reanalysis data can capture due to the relatively coarse spatial resolution and the impact of heat waves is actually larger than thought. For example, there is evidence in other high-elevation zones that shortwave radiation plays a strong role in driving melt, even in sub-zero temperatures (Matthews et al., 2020), that could also explain some of the mismatches we see here. This summer, an automatic weather station near Gore

at around 3900 m a.s.l. was installed by the colleagues from the University of Milano which will enable us to better monitor the local weather and to better link it with surface velocity and runoff in the future. Unfortunately, we don't have any higher weather stations (yet) to be able to analyze how much the weather drives melt higher on the glacier. As there is only a limited amount we can say about this, we will not include it into the manuscript and hope for your understanding.

Reference: Matthews, T., and Coauthors, 2020: Going to Extremes: Installing the World's Highest Weather Stations on Mount Everest. Bull. Amer. Meteor. Soc., **101**, E1870–E1890, https://doi.org/10.1175/BAMS-D-19-0198.1.

● Revisit the trends from the results and highlight what is (un)expected, novel, and specific to debris-covered glaciers. How do the patterns seen here depend on the debris cover?

Great idea! We included our highlights and findings in the new section 5.3 with the title "Impact of debris cover" (Line 373):

"Debris covered glaciers show a different response to global warming, because ice melt is reduced by the insulating effect of the debris cover over extensive part of the ablation area (in case the debris reaches a critical thickness). Another characteristic of debris-covered glacier is their ability to store large amounts of water in supraglacial lakes, which are suddenly released when crevasses opens resulting in basal sliding.

The important findings of this study are 1) the period of several days with positive air temperatures in spring lead to an increase in lake rise and to a snowmelt up to 4700 m a.s.l., 2) the large melting area of snow and ice causing the spring speed-up and the high summer glacier velocities, and 3) the fact that the Baltoro glacier also shows an acceleration in fall, caused by the drained supraglacial lakes. The area affected by the melt and its effects on glacier dynamics is therefore surprisingly large. Despite the insulation due to the debris cover on the main branch of the Baltoro Glacier, the ice cliffs and supraglacial lakes act as hotspots that react sensitively to the rise in temperature and lead to an additional ice ablation."

Section 5.1 goes through an overview of the inferred hydrology of Baltoro Glacier. The presented distributed/channelized switch conceptual model is well understood, but I wonder if in this case there might be more going on. For example, might changes in bed connectivity (e.g., Andrews et al., 2014; Hoffman et al., 2016) or a more nuanced understanding of basal hydrology and sliding (e.g., Gilbert et al., 2022) also explain some of the inferred patterns? What is the basal environment (hard bed, soft bed?), and might storage/drainage through till be important here? A similar comment could be made about the description near line 30.

Many thanks for the references. It is an interesting point that the connectivity varies spatiotemporally and indeed it would be good to know if this phenomenon exists as well at the Baltoro Glacier. The mentioned references are using borehole and GNSS measurements to get a high temporal resolution of in-situ data. Though, the glaciers in the Karakoram have not been sufficiently studied in the past due to their remoteness and the lack of in-situ data makes a better analysis of their behavior very difficult. Our plan was to conduct a field campaign to investigate the hydrology in more detail, unfortunately we did not receive the permission from the Pakistan government – just to show the dilemma and to make it clear why we cannot explain small-scale phenomena.

We assume that the glacier bed is soft as there is a lot of debris available and a lot of loose material in front of the glacier. As this is just an indirect assumption, we refrain to add this information in the manuscript. We hope that you can understand our point of view.

I think it would help to add one or two sentences about how you infer sliding from surface velocities (perhaps in Section 5.1, to support some of the conclusions). I imagine you are assuming the deformational component does not change significantly through the year, and so the only remaining mechanism to explain the seasonal velocity variations is a change in basal velocity. Making this explicit would support the later conclusions nicely.

Good idea! We added the following sentence in Line 348: "A notable indication of basal sliding during the summer months is the substantial increase, ranging from 2 to 6 times, in velocity compared to the winter (Mayer at al. 2006)."

What controls the timing of lake drainage? Do lakes reach a threshold size and then drain, or is it more complex? See also line 327 comment.

We assume that this is due to the water volume (and the underlying drainage system) which we unfortunately cannot estimate by means of Earth observation data.

Line 324: I am curious if it's possible to use the difference between snowmelt and the runoff index to try to quantify, or at least observe, a release of stored water when the drainage system connects and channels can evacuate the system. This is hinted at here, but it would be interesting to look into this in a little more detail.

Good idea! However, since we can derive only the area of snowmelt and runoff and not the volume, this estimation could result in large uncertainties leading to wrong assumptions. Therefore, we refrain from following up on this.

Line 327: The idea that high summer velocities enable crevasse development and lake drainage has some nuance and subtlety (e.g., Poinar & Andrews, 2021). Have you looked at this hypothesis by estimating strain rates (as best as possible with the limited resolution from remote sensing data) to see if this is at least reasonable? Further, wouldn't this mechanism predict lakes to drain mostly during acceleration, whereas Figure 2 shows lake area sometimes decreasing while velocities also decrease? I do think this is a reasonable mechanism to propose, but some more care should be taken here.

Thank you for the interesting idea and reference. Indeed, it would make sense to have a look into the stress rate and to analyze whether there is a connection with the lake drainage. However, in our opinion it would be out of scope of this manuscript. Nevertheless, we will keep it in mind for our next analyses.

Line 368: It would be interesting to see the difference between peak surface velocities and net surface displacement over the melt season and over the calendar year (especially as the authors have elsewhere cited Sundal et al., 2011). This would be a nice addition to the discussion, then could be referenced here.

Thank you so much for the suggestion! The displacement increases in m d$^{-1}$ as well as in percent are given in discussion and conclusion, which effectively show the evolution of the velocities through the melt season as suggested. In the conclusion, we added the displacement increases as follows (Line 392): "The snowmelt of the tributary glaciers, covering up to 64 % of the total area of Baltoro Glacier (Fig. 3b) and lasting from April to November/December (Fig. 2), had the greatest impact to the basal sliding that led to the spring speed-up of 0.1-0.16 m d$^{-1}$ (30-35%) and the summer velocity increase by 0.2-0.3 m d$^{-1}$ (75-100%, Tab. A1 and A2). The subsequent transition from inefficient to efficient

drainage lead to a glacier slow down. Afterwards, the supraglacial lakes with a summarized area of 3.6-5.9 km$^2$ (Fig. 3b) contribute to the fall speed-up which has a lower magnitude by 0.1-0.2 m d$^{-1}$ (20-30%) than the summer velocity peak."

**Technical corrections**

Line 28: Does Baltoro Glacier have basal motion in winter? I would tend to think of basal motion as a continuum, and not that basal sliding initiates from zero over winter, but rather that basal motion accelerates as effective pressure decreases. For example, this helps to explain why there are winter variations in velocity, otherwise the deformational component would have to be changing rapidly.

This is a good question! We assume as well that the winter surface velocity is mostly driven by basal sliding, but this is difficult to prove. As there are only a few in-situ data of the Baltoro Glacier, the estimated global ice thickness could be used to calculate shear stress and thus deformation rates. However, the ice thickness, which is estimated from surface velocity derived from Earth observation data (Welty et al., 2020), is subject to a large uncertainty, which in turn would lead to a large error propagation. For this reason, we refrain from making statements regarding basal sliding in winter. In summer, however, basal sliding clearly takes place due to the significant increase in glacier velocity.

Reference:

Welty, E., Zemp, M., Navarro, F., Huss, M., Fürst, J.J., Gärtner-Roer, I., Landmann, J., Machguth, H., Naegeli, K., Andreassen, L.M., Farinotti, D., Li, H., and GlaThiDa Contributors (2020): Worldwide version-controlled database of glacier thickness observations. Earth System Science Data 2020. DOI: https://doi.org/10.5194/essd-2020-87

Line 30: I do acknowledge that there are different ways to explain subglacial drainage evolution. However, to some readers (especially modellers), "inefficient channels" is an unusual phrase. If the authors agree, consider changing to describe these channels as small, incipient, etc.

Thanks for the explanation. We corrected it to "small and incipient subglacial channels" (Line 28).

Line 33-35: "In the absence of meltwater the ice- overburden pressure is larger and ...". Would this be more clear to say that the effective pressure is larger (because water pressure decreases, not because ice overburden changes)?

Corrected to "effective pressure is larger" (Line 31).

Line 35: Is there evidence that regelation contributes to channel closure?

We added the reference of Weertmann et al. (1957) (Line 31): "In the absence of meltwater the effective pressure is larger and leads hence to a closure of the channels through regelation (Weertman, 1957) and creep (Benn et al., 2019; Flowers, 2015; Jiskoot, 2011)."

Line 35-39: I prefer to think that glaciers respond to the balance between driving and resistive stresses. Driving stress should vary slowly, since this is controlled by geometry. The resistive stress (especially basal traction in response to effective-pressure variations) instead can vary over short timescales.

Thanks for the hint. It seems that I have lost my thought while writing. The explanation was corrected as follows (Line 33): "The driving stress is influenced by gravitational acceleration, ice density, thicknesses and surface slope and varies slowly. The resistive stresses arise from the drag at the glacier and by dynamical flow resistance. Glacier flow processes can be divided into ice deformation, basal sliding, and sediment deformation."

Line 37: I wonder if maybe the 50% contribution of sliding to total glacier surface velocity has been taken out of context. For Greenland outlet glaciers, or surge type glaciers, basal motion can contribute well over 50% to velocity.

You are right - as there is no field data for the glaciers in the Karakoram to support this statement, it would be better to delete the specific percentage (Line 35): "Sliding and bed deformation occur only in the case of temperate and polythermal glaciers with a higher contribution of basal sliding than internal deformation velocities (Boulton and Hindmarsh, 1987; Jiskoot, 2011)."

Line 48: Would it also be conceptually possible for supraglacial lake discharge to contribute to enhanced subglacial channel development, eventually reducing ice surface velocities?

You are right – with an inefficient drainage system, the discharge can lead to high surface velocities and with an efficient drainage system to low velocities. We corrected the sentence as follows (Line 45):

"In an inefficient drainage system, supraglacial lake discharge can support basal sliding and hence cause higher glacier velocities (Sakai and Fujita, 2006; Sakai, 2012; Watson et al., 2016; Benn et al., 2017; Miles et al., 2020). In contrast, the supraglacial lake drainage could lead to a transition from an inefficient to an efficient drainage system and hence lead to a slowdown of the glacier velocity (Vincent and Moreau, 2016; Stevens et al., 2022)."

Line 49: This is a long (but nicely detailed) paragraph. It might be easier for the reader to split into two paragraphs, separating background theory from details specific to Baltoro Glacier.

Good suggestion! Done.

Line 74: What proportion of the debris covered area is enhancing vs. inhibiting surface melt?

That would indeed be interesting to know. The debris-cover thickness increases with the height elevation and additionally spatially due to advection, debris and meltwater movement, and slow cycles of topographic inversion. Hence, the debris coverage is spatially very heterogeneous. Mihalcea et al. (2006) concluded that the "debris cover reduces the ablation over a large part of the glacier", though they could not determine the exact proportion based on their stake measurements. This question can probably only be answered with UAV thermal infrared images.

We added the two sentences (Line 77): "Furthermore, the debris thickness varies spatially due to advection, debris and meltwater movement, and slow cycles of topographic inversion (Nicholson et al., 2018; Huo et al., 2021) which impedes the determination of the area proportion enhancing surface melt."

Reference: Mihalcea, C.; Mayer, C.; Diolaiutt, G.; D'Agata, C.; Smiraglia, C.; Lambrecht, A.; Vuillermoz, E. and Tartari, G.: Spatial distribution of debris thickness and melting from remote-sensing and meteorological data, at debris-covered Baltoro Glacier, Karakoram, Pakistan. Ann. Glaciol., 48:49–57, 2008. doi: 10.3189/ 172756408784700680.

Figure 1: Adding elevation labels, at least for the boundary between regions, and perhaps distance markers (as long as the figure does not become too busy) would help readers who are not familiar with this glacier.

We added elevation and markers for the four locations (Urdukas, Yermanendu, Gore, and Concordia) and distance markers. We hope that helps the readers to better understand the region.

[Figure]

Section 3.2: I understand that this method is detailed in a previous paper, and the description here is in appropriate detail. Can you very briefly comment on the sensitivity/performance of the different sensors used here?

Sure! We added following information (Line 122): "The advantage of the multi-sensor approach is that it highlights the strengths of each sensor and compensates for their weaknesses: Sentinel-2 provides a continuous, radiometrically stable time series with a temporal resolution of 12 days, which is filled by the high temporal sampling of PlanetScope data. During periods of cloud cover SAR data provide important information. The disadvantages of the SAR data, i.e. lake area underestimation and missing data from side-looking radar geometry and undulating glacier surface topographies, can be compensated by using optical data acquired on the same day"

Section 3.3: The term 'runoff-index' used on line 123 is a great way to acknowledge the simplified metric used here (as compared to runoff in units m3 s -1 ). I was confused about the type of runoff being measured until this section. Could identify that you are measuring proglacial runoff (i.e., as opposed to runoff through supraglacial streams) and briefly acknowledge the area index the first time you say 'runoff' (including abstract)?

Thanks for the suggestion. We added following explanation to the abstract (Line 6): "The runoff-index is defined as the estimated surface areal coverage of the proglacial stream given in km$^2$ and is used as a proxy of the quantitative runoff." and in the introduction (Line 64): "runoff-index estimated as surface areal coverage of the proglacial stream given in km$^2$ used as a proxy of the quantitative runoff."

Section 3.5: Since I am not familiar with the HAR data specifically, I found the hierarchy between the different levels of meteorological data confusing here.

Both data are now mentioned at the beginning of the section (Line 173): "The near surface air temperature at 12 noon and total precipitation data (daily sum) were obtained from the daily interpolated HAR data set (version 2) provided by the Chair of Climatology, TU Berlin. The air temperature data (2 m height) from an automatic weather station located at Urdukas (Mihalcea et al., 2008) was needed for the local downscaling of the HAR data."

Line 179: What proportion of lakes did not drain or fill during the observation period?

All the lakes are affected by filling or draining. During the first observation period (13 April to 7 May), all the supraglacial lakes fills up (1.1 m$^2$). The second period (7 May to 18 July) is characterized by filling

from 7 May to 2 July by 1.3 m$^2$ of all lakes, followed by filling (0.33 m$^2$) and simultaneous draining (0.46 m$^2$) from 2 July to 18 July. This information was added in Line 289.

Line 185: Is the end of the ablation season defined as 30 September based on the water year starting on 1 October, or was 30 September determined to be the usual end of summer melt?

End of September is usually the end of summer melt. We changed the sentence to "…the second period from the time of maximum glacier velocity until the end of the snowmelt season on 30 September…" (Line 203).

Section 3.8: Do you see lakes drain fully within the 2–4 day observation frequency? If so, does the interpolation artificially smooth this lake area signal, i.e. turn a discrete drainage event that happened at some time in the 2-4 day window into smooth linear decrease at daily resolution?

Good point! The PlanetScope data with a temporal resolution of 2-4 days show that some lakes drain within 2-4 days (but not all of them). In this case, the exact date of the duration of the drainage cannot be determined with Earth Observation data and the linear interpolation does not represent the real drainage behavior. We were aware that the signal was being artificially altered, but decided to use the linear interpolation anyway as the lakes show different drainage behaviors which is difficult to represent individually and for reasons of simplification.

Table 1: I appreciate the level of detail in this table. What is the difference between "aggregate" and "cumulative" lake area? And are these areas presented just for active lakes?

You are right, the two terms are ambiguous. The idea was to distinguish between the maximum lake area for each summer and the summarized area of the entire summer season. For a better discrimination, we now use the terms "maximum area" and "summarized area". Both areas refer to permanent and seasonal lakes.

Line 211: What factors do you think cause the hardly discernible time delay from lower to upper sectors here?

The change from winter to spring happens very abruptly, and the warmer temperatures also reach higher altitudes. We added the following explanation (Line 239): "A temporal delay from the lower (western) sector to the upper (eastern) sector was hardly discernible which could probably be explained by an abrupt transition from winter to spring with higher air temperatures at higher altitudes."

Figure 2 is an excellent comparison of so many fields of data. Is there a way to also label each year with its total positive degree days without making the figure too busy?

You are right, adding the positive degree days into the plots could deliver interesting information. Though, they have a different value range and a third x-axes would be needed, which makes the plot too confusing.

L321: Should this be "Schoof, 2010"?

You are right, his family name is Schoof. Thank you for spotting this.

L328: "The drained lake which is only disposable after the slowdown" is unclear.

We changed the sentence (Line 359) to "The water of the lakes draining after the slowdown lead, in addition to the snowmelt, to a water surplus and hence to an increase in water pressure and basal sliding during fall (Fig. 2, Tab. 2, and Tab. 3), but with lower magnitude by 0.1-0.2 m d$^{-1}$ (20-30%) than the summer velocity peak (Hewitt, 2013; Hart et al., 2022; Nanni et al., 2023)."

**References**

Andrews, L. C., Catania, G. A., Hoffman, M. J., Gulley, J. D., Lüthi, M. P., Ryser, C., ... & Neumann, T. A. (2014). Direct observations of evolving subglacial drainage beneath the Greenland Ice Sheet. Nature, 514(7520), 80-83.

Gilbert, A., Gimbert, F., Thøgersen, K., Schuler, T. V., & Kääb, A. (2022). A Consistent Framework for Coupling Basal Friction With Subglacial Hydrology on Hard-Bedded Glaciers. Geophysical Research Letters, 49(13), e2021GL097507.

Hoffman, M. J., Andrews, L. C., Price, S. F., Catania, G. A., Neumann, T. A., Lüthi, M. P., ... & Morriss, B. (2016). Greenland subglacial drainage evolution regulated by weakly connected regions of the bed. Nature communications, 7(1), 13903.

Poinar, K., & Andrews, L. C. (2021). Challenges in predicting Greenland supraglacial lake drainages at the regional scale. The Cryosphere, 15(3), 1455-1483.

**Review 2:**

Overall, this paper presents some interesting patterns concerning intra-annual and inter-annual variations in the velocity of Baltoro Glacier, and their connections to surface hydrology (particularly in relation to snowmelt and supraglacial lakes). I find that the start of the paper is relatively well developed, but more detail is needed in the Results, and much more work is needed in the Discussion and Conclusions. A particular issue with the latter sections is that large blocks of text are presented in single paragraphs, with little apparent connection between adjacent sentences, which makes it difficult to understand what the authors are trying to say. I also find that there are large jumps in logic (e.g., L331-L337), but currently insufficient analysis and presentation of the data to back them up. I'm not saying that the interpretations are incorrect, but rather that the results need to be better presented and analyzed, and perhaps additional figures included, to make the arguments more convincing. In addition, there is a lack of specificity and detail throughout the manuscript, particularly in relation to the methods. For example, little mention of the resolution of any input data is provided.

Finally, the paper needs to make better reference to existing literature, particularly in the Discussion. For example, Quincey et al. (2009) investigated climate-velocity relationships on Baltoro Glacier, but isn't discussed. Dehecq et al. (2019) also looked at long-term trends in glacier velocity in the Himalayas, including over the study area, yet isn't mentioned.

Detailed comments are provided below, several of which relate to the points above.

Dear Reviewer,

thank you very much for the useful comments regarding our work and especially all the suggestions. They helped a lot to improve the manuscript. Accordingly, parts of the sections in results and discussion have been completely rewritten and revised in order to carve out our findings and to link them better with our figures and tables. We deeply appreciate your conscientious review.

Your reviews have been answered line by line. For a better readability of our responses, the answers to the reviews and the corrections in the manuscript are shown in blue.

Kind regards,

Anna Wendleder with co-authors

Title: include Pakistan in the title? Just in case a reader isn't familiar with Baltoro

Thank you very much for this suggestion. The title has been changed to "Basal Sliding and Hydrological Drainage at Baltoro Glacier, Pakistan"

Abstract: the abstract is long, which makes it a bit difficult to ascertain the most important findings from your study. Cutting or shortening the following sentences, which provide background information rather than new results, should make it easier to follow:

"Surface meltwater directly influences glacier velocity, as liquid water at the bed allows the glacier to slide. However, prolonged discharge of water at the bed increases the efficiency of the drainage system and decreases the amount of sliding. Due to the presence of an insulating debris mantle, debris-covered glaciers respond in a more complex way to changes in climate than those that are debris-free. The influence of long-lasting high temperatures on melt processes and, subsequently, supraglacial lake formation, and 5 the triggers of basal sliding have not yet been sufficiently analyzed and understood."

And: "The relationship and dependency between the variables were examined with Pearson correlation and linear regression, respectively. Additionally, the temporal delay between snowmelt peak and glacier flow acceleration was determined."

Following your suggestions, the first part of the abstract has been shortened (Line 1): "Glacial surface meltwater directly influences glacier dynamics. However, in the case of debris-covered glaciers, the drivers of glacier velocity and the influence of supraglacial lakes have not yet been sufficiently analyzed and understood."

The second mentioned sentence has been deleted.

L12 & L18: Include percentage change for the speed-ups that you describe, in addition to m d-1

That's a good idea to make the changes comparable to other studies! Therefore, we have added the relative change: "...with an increase in summer by 0.2-0.3 m d$^{-1}$ (75-100%) ..." (Line 9) and "… a lower magnitude by 0.1-0.2 m d$^{-1}$ (20-30%) …" (Line 14).

L29: I'm not sure that the statement 'the glacier lifts up' is always applicable for glacier accelerations. For example, if increased water pressure at the glacier bed leads to a reduction in the strength of subglacial sediment, then an acceleration could presumably occur without associated uplift.

Thank you for the correction. We have changed the sentence as follows (Line 23): ". This influx of meltwater into the subglacial drainage system leads to an increase in basal water pressure. When subglacial water pressure approaches ice-overburden pressure, basal traction decreases and sliding is initiated as the ice decouples from the bed. The glacier - in case of a hard glacier bed - lifts up and accelerates (Weertman, 1964; Lliboutry, 1968; Iken and Bindschadler, 1986; Nolan and Echelmeyer, 1999; Sugiyama et al., 2011; Hoffman et al., 2016; Benn et al., 2019)."

L30: change to 'inefficient subglacial channels' and L32: change to 'efficient subglacial channel' to clarify that you're referring to basal channels here (e.g., as opposed to supraglacial or englacial)

Thanks for the hint. As the other reviewer commented that "unefficient channel" is an unusual phrase for modelers, we have corrected it to "small and incipient subglacial channels" (Line 28). Your second comment has been included in Line 29: "In an efficient subglacial channel system…"

L34: delete 'hence'

Deleted

L37-39: the statement that basal sliding and bed deformation can contribute up to 50% of the total glacier surface velocity isn't well supported by field measurements. Although there are few detailed in situ studies, the few available measurements suggest that bed velocity accounts for between ~70% (Raymond, 1971; Willis at al., 2003) and 99% (Seroussi et al., 2011) of surface velocity, and an assumption of 90% to 100% is often used to compute depth-averaged velocity from surface velocity, particularly for large glaciers. See last paragraph in section 2.3 of Kochtitzky et al. (2022) for a detailed discussion and associated references.

Thank you for the reference and the information. We fully agree with the reviewer: as there are no field data for the glaciers in the Karakoram to support this statement, we have deleted the specific percentage (Line 36): "Sliding and bed deformation occur only in the case of temperate and polythermal glaciers with higher contribution of basal sliding than internal deformation velocities".

L43: change 'widely been' to 'been widely'

Corrected.

L43: change to 'increase of air temperatures…'

Corrected.

L51: reference to Quincey et al. (2009) would also seem to be relevant here since this was written by some of the co-authors of the current study, and used SAR data to find some similar velocity patterns for Baltoro Glacier

We agree. The reference has been added here.

L83: irregular weakening of what? (e.g., westerlies? Tibetan Anticyclone?)

Changed to "In the case of an irregular weakening of the Tibetan Anticyclone" (Line 86).

Fig.1: throughout the text you refer to features present at different elevations (e.g., snow above 5400 m: L88), but don't provide elevations in any of your figures. It would therefore be useful to provide this: e.g., as contours added to Fig. 1, or a separate figure with a DEM of the region.

Thank you for pointing this out. Therefore, in Figure 1 and 3, we have added the elevation of the four locations (Urdukas, Yermanendu, Gore, and Concordia). In addition, we have used the DEM as background image for Figure 3 to better present the topography of the area and referred explicitly to the elevation of the snow melt in chapter 4.2.

L90: change 'chapter' to 'section'

Corrected.

L94: provide the resolution for the input Sentinel-1 data. Also a Table with the dates and IDs of the images that you used to derive the velocities (if the Table is long, then it could be put in supplementary material)

We agree that a further specification is needed here. Therefore, the spatial resolution of Sentinel-1 has been included in Line 97: "The glacier surface velocity was calculated from the Sentinel-1 Interferometric Wide Swath Single Look Complex (SLC) data with a spatial resolution of 10 m." The table with acquisition dates and absolute orbit number has been included in the supplementary material (Table A3).

L96: which software or code did you use to process the velocity data? E.g., GAMMA?

The processing was indeed done using GAMMA. We have added this information (Line 100): "Feature tracking and subsequent processing steps were implemented in GAMMA (release 20211201, Wegmüller et al., 2016)."

L100: describe how the accuracy of the velocity maps was determined

We agree that this aspect needs further clarification. We have thus added the following information (Line 104): "The mean accuracy of the velocity maps is 0.06 m $d^1$ of a pixel resulting in a standard

deviation of 0.042 m d[1] which is calculated according to the formula in Strozzi et al. (2002) and Friedl et al. (2021)."

L102: add markers to Fig. 1 (or add a centreline with distance markers) to show exactly where these points are located

As suggested by the reviewer, we have added markers for the four locations (Urdukas, Yermanendu, Gore, and Concordia), labels for the elevation and distance markers. We hope this improves the readability of the paper.

[Figure]

L105: provide the resolution of the images that you used. You state the final spatial sampling is 10 m on L117, but it's useful for the reader to also understand what the resolution of the source data was.

Thank you very much for the hint! We have added the spatial resolution of each sensor (Line 138): "Sentinel-2 Multi Spectral Instrument (MSI) orthorectified Level-1C Top-Of-Atmospheric products (spatial resolution of 10 m) were… The PlanetScope Analytic Ortho Scene Products (Level 3B, spatial resolution of 3 m)) were … Sentinel-1 Interferometric Wide Swath Single Look Complex (SLC, spatial resolution of 10 m)) C-band and TerraSAR-X ScanSAR Multi-Look Ground Range Detected (MGD, spatial resolution of 40 m)) X-band data were…."

L124: I don't understand what 2.7 km2 area refers to here. The white box in Fig. 1 seems to have sides of ~4 km, so an area of ~16 km2. Or perhaps you're only referring to the area occupied by channels inside the white box? If so, this needs to be specified.

You are right, it is misleading that the white box specifies a larger area around the runoff than the defined 2.7 km$^2$. We have changed the bounding area in Figure 1 accordingly (see above).

L125: you refer to 'water-filled channel area', but provide linear units (in m). For area these should be in m2, or change the wording.

We apologize for the mistake. It should mean "water-filled channel width". We have corrected it in Line 137.

L126: similar to comments on previous sections, provide the resolution of your input data

We have added the spatial resolution for Sentinel-2 and PlanetScope (Line 138): "We used the Sentinel-2 Multi Spectral Instrument (MSI) orthorectified and MAJA atmospherically corrected L2A products (spatial resolution of 10 m) and the PlanetScope Analytic Ortho Scene Products (Level 3B) Surface Reflectance (SR) data (spatial resolution of 3 m)."

L135: you refer to the 'manually digitized reference', but don't specify what this is or how it was chosen or analyzed

Thank you for pointing this out. We have corrected the sentence as follows (Line 147): "To evaluate the accuracy, four reference data sets with different acquisition dates of the run-off area were digitized manually on the basis of the near-infrared band."

L137: to avoid ambiguity, include months to define the 'beginning of the ablation season' and 'during the ablation season'

This is an excellent suggestion. The months have been added (Line 149): "The time series has a temporal sampling of 5-15 days in the months of May to June, which is characterized by a higher cloud coverage and 1-5 days in the months of July to September with less cloud coverage."

L140: when you refer to 'complete Baltoro Glacier', does this mean the entire outlined area in Fig. 1? If so, then specify this. As with earlier comments, also provide resolution of input data.

Both points have been added in Line 153: "We mapped the snow and ice melt on the complete Baltoro Glacier (see outline of the three polygons in Figure1) which includes the debris-covered and debris-free part of the glacier from the Sentinel-1 Interferometric Wide Swath C-band data with a spatial resolution of 10 m."

L149: provides the elevation values that are defined by the 10% and 90% percentiles (and related to earlier comment for Fig. 1, it will be useful to have an elevation plot so that the reader can understand where these are)

Please excuse the misunderstanding. The elevation values are changing during the year and therefore cannot be specified here in detail. In order to clarify this aspect, we have changed the sentence as follows (Line 164): "By intersecting the wet snow mapping with the Copernicus DEM (1 arc second, version 2022_1, (ESA, 2019)), the 10 % and 90 % percentile of the elevation of the aggregated snowmelt area are shown, which changes over time."

L155: temporal delay in what? E.g., in seasonal surface melt patterns?

We apologize for being unclear on this. Yes, we meant the snowmelt pattern. To clarify it, we have changed to sentence as follows (Line 170): "As area and elevation of the glacier increases from west to east, the temporal delay of the snowmelt in the three sectors reflects the vertical gradient."

L159: please define which temperature(s) you're looking at. E.g., mean daily? Daily maximum? Daily minimum?

We took the air temperatures at 12 noon. This has been specified in more detail (Line 173): "The near surface air temperature at 12 pm (noon) and total precipitation data (daily sum) were obtained from the daily interpolated HAR data set (version 2) provided by the Chair of Climatology, TU Berlin."

L160: please provide the resolution (grid cell size) to which this dataset was downscaled

Apologies for the unclear description. We have corrected the sentence and added the spatial resolution (Line 177): "The global European Centre for Medium-Range Weather Forecasts (ECMWF) Re-Analysis (ERA-5m, spatial resolution of 80 km) was dynamically downscaled by regional climate models for the High-Mountain Asia to produce the regional refined HAR data set with a spatial resolution of 10 km."

L166: provide more detail about the Urdukas automatic weather station: e.g., did this record overlap the entire period of the HAR data set? Also more detail about the comparison: e.g., did the lapse rates vary by month/season, and if so were different values used for different parts of the year?

We have included more details about the automatic weather station (Line 174): "The air temperature data (2 m height) from an automatic weather station located at Urdukas (Mihalcea et al., 2008) was needed for the local downscaling of the HAR data. The weather station is equipped with an air temperature sensor (thermo-hygrometer with radiation shield) taking hourly measurement for 2011."

Furthermore. the explanation about the lapse rate calculation has been added in Line 183: "To downscale the air temperatures, a mean lapse rate for each month was determined based on the air temperatures data from the automatic weather station using a fitting curve (3rd order). The derived lapse rates varied between 7° C km$^{-1}$ (December) and 11° C km$^{-1}$ (June)."

L175: this is an interesting way of plotting the data, but one that I'm not particularly familiar with. Can you provide reference(s) to studies that support the use of this method?

We appreciate your suggestion. We have added the references of Sahr (2011) and Brodsky (2018) explaining the hexagonal grid in detail and Fichtner et al. (2023) giving a nice application example.

L183: I assume that you're referring to the r value here? If so, then change 'Pearson correlation' to 'Pearson correlation coefficient (r)' to avoid any potential ambiguity.

Yes, we do. To avoid ambiguity this has been changed accordingly.

L199: similar to comment for L159, define which temperature you're talking about here. E.g., mean daily? Daily maximum?

We refer to the daily 12 noon temperature. This information has been added (Line 219): "daily noon air temperatures"

L206: change 'started on' to 'started between'

Corrected.

L211: I don't really follow the wording here: 'snowmelt area... covering only the debris-free part of the glacier.' This makes it sound as if melt only occurred on the debris-free part of the glacier, but the debris-covered part of the glacier is at lower elevation and has lower albedo, so shouldn't melt have been occurring there as well? Or perhaps no snow was present there? This needs to be clarified.

Thank you for pointing this out. We have rewritten the sentence as follows (Line 238): "In July 2019, the total snowmelt area had a maximum of 353 km$^2$ and also covered the debris-free part of the glacier."

L214: change to 'maximum total area'

Corrected.

L215: define which peak you're referring to at the end of this line (e.g., peak in total lake area?)

Thank you, we have changed the sentence to better define our explanation (Line 229): "The formation of the lakes started early-April to early-May and reached the peak of the maximum total area from end-May/early-June (2020, 2022) through mid/end-July (2017, 2018, 2019) to mid-August (2021)."

L222: here and elsewhere it would be useful to provide the % change in velocity, in addition to the absolute change that you already list

We have added the relative velocity change in results and discussion.

L218-L235: I find it hard to follow the text here as there's so much detail provided for different locations, years and seasons all in the same para, with the timing of some descriptions unclear (e.g., L234: which year(s) does the statement 'After the summer peak' refer to?). To make the text easier to follow I suggest breaking it up into smaller paragraphs (e.g., organized by season or year), and/or adding another figure (e.g., using a Hovmoller plot, similar to Fig. 3 in https://doi.org/10.1029/2021GL097085)

We appreciate your assessment and thank you for the suggestions. The section "4.1. Temporal relationship" has been restructured to focus more on relationships and therefore on a specific season (please see revised manuscript). The statement 'After the summer peak' has been removed during the revision. The paragraph explaining the results of the glacier surface velocity has been divided into the individual subsubsections and revised for better readability (Line 215). The revised text reads as follows:

"4.1.2 Spring and summer speed

Between May and September, the melt of snow and ice fluctuated gradually through time between minimum and maximum extent, reaching spatial maxima of 90 km$^2$ for the western sector (2018), 84 km$^2$ for the central sector (2017), and 190 km$^2$ for the eastern sector (2019). In July 2019, the total melt area had a maximum of 353 km$^2$ and also covered the debris-free part of the glacier. It corresponds to 64 % of the total area of Baltoro Glacier. A temporal delay from the lower (western) sector to the upper (eastern) sector was hardly discernible which could probably be explained by an abrupt transition from winter to spring with higher air temperatures at higher altitudes. Differences only existed in the melt area as the east sector covers a larger glacier area than the western and central sector. The first and second melt event occurred in the same period as the spring speed-up and lake evolution. In the area upwards of Urdukas which is characterized by less debris cover, the meltwater accumulates in the supraglacial lakes and drains into the subglacial system or flows laterally into larger streams. In the area downstream of Urdukas, there are no larger supraglacial channels and the water can drain through moulins and crevasses to the ground.

The winter surface velocities (1 October until the first date with positive air temperatures usually at the beginning/end of March) ranged over the years between 0.05 and 0.37 m d$^{-1}$ and were followed by a spring speed-up. This event lasted usually from the end of April to the beginning of May and led to a relative increase of 0.1–0.16 m d$^{-1}$ (30-35%) above Yermanendu. In 2017, 2019, 2020, and 2021, the velocities at Gore exceeded even those at Concordia at this time. The spring speed-up was particularly visible in 2018 affecting all four locations. However, in 2022, the velocities at Urdukas were rarely impacted. After the spring speed-up, the flow at Concordia accelerated continuously until the summer peak of 0.46-0.59 m d$^{-1}$ was reached in June to July. Only in 2022, a short slow-down between the speed-up and the summer peak was recognizable. The high lasted for 11-12 days in 2017, 2019,

2021, and 2022 and 22 days in 2018 and 2020. However, not all four locations experienced the maximum velocity at the same time: in 2022, the summer peak affected all four locations; in 2021, only the area upglacier of Yermanendu was influenced; in 2020 and 2019, the velocities downglacier of Gore reached their maximum during the spring speed-up; in 2018, the velocities at Concordia and Gore had their high during summer peak, Yermanendu and Urdukas during spring speed-up; in 2017, the velocities upwards of Yermanendu reached their maximum during summer peak and Urdukas afterwards. Additionally, the year 2021 was characterized by a continuous transition of spring speed-up and summer peak. Between June and July, the surface velocities between Concordia and Urdukas spread with a difference of up to 0.46 m d$^{-1}$ with an increase at Concordia and decrease at Urdukas. In 2017 and 2021, this phenomenon happened after the summer peak and in 2018-2020 between speed-up and summer peak."

We hope that the reconstruction helps to better understand the chapter and that an additional representation according to Hovmöller is not necessary.

Table 1: for the supraglacial lakes, what's the difference between aggregated area and cumulative area?

Thank you very much for spotting this, the two terms are ambiguous indeed. The idea was to distinguish between the maximum lake area for each summer and the summarized area of the entire summer season. For a better discrimination, we now use the terms "maximum area" and "summarized area" and have updated the manuscript accordingly.

L236-L242: this para is mainly an interpretation of the results, rather than presenting new information, so would seem to fit better in the Discussion section than the Results section.

You are right. We have shifted the sentences to the discussion as suggested.

L252: Figure 3 would be easier to follow if you could label the location of Urdukas, Gore, Concordia and Yermanendu on it (perhaps place the labels over adjacent bedrock, so that they don't cover the colours for the velocities). I also find the velocity colours a bit difficult to distinguish on this figure, particularly for the light shades used for low velocities (white in particular for 0.00-0.20, which makes it almost impossible to distinguish against the hillshade).

Thank you for this suggestion. Figure 3 has been renewed. It includes now the locations of Urdukas, Yermanendu, Gore, and Concordia. The colours of the glacier velocity have been changed to better discriminate them from the supraglacial lakes. Additionally, the outline of the supraglacial lakes has been intensified.

[Figure]

L256: label where Mandu Glacier is on a figure

Thank you for the hint. To reduce the names and thus the information in Fig. 1, we have changed the sentence (Line 287): "In the second period (Fig. 3b), the higher velocities of 0.4- 0.6 m d$^{-1}$ expanded up to 5 km below Yermanendu."

L260-L282: there are lots of details provided in this para, which makes it difficult to follow and to understand what the most important relationships are. To start, I suggest breaking it up into two paras for the two periods. Second, be precise about the level of significance of your correlations, and perhaps only refer to ones that are statistically significant (e.g., $\rho > .05$). At the moment you variously describe correlations as being 'strong', 'high', 'low', 'significant positive', etc., but don't define what these terms mean exactly.

Thank you for your suggestions. All your comments have been implemented. The section has been completely rewritten and focuses now on the differences between both periods to make it easier to read. Variables that are not statistically significant were not mentioned in the text and the adjectives describing the correlation were reduced and better defined (Line 295). The revised text reads as follows:

"In the two periods, the correlation of the variables temperature and melt (0.7-0.95) as well as temperature and supraglacial lake (0.58-0.89) showed a similar pattern with a high relationship (>0.6) though with negligible minor changes in the Pearson correlation during both periods (0.1-0.2); except for the decrease by 0.4 for temperatures and supraglacial lakes in the second period of 2020. A high degree of correlation existed between melt and runoff-index (0.65–0.93) and supraglacial lakes (0.63–0.94), respectively, albeit with outliers in both relationships in the second period of 2020 (decrease by 0.4 and 0.9). The relationship between supraglacial lakes and velocity as well as melt and velocity showed the same pattern with higher correlations at Concordia and Yermanendu and lower correlations in Urdukas in both periods. However, the correlations of the second period vary significantly (0.5 to -0.8) over the years and generally decrease by 0.4-1.3 in the second period. Particularly noticeable was the large decline (<1.5) in Urdukas in both parameters. The correlation

between velocity and runoff decreased by to 1.3 in the second period. However, compared to the two pairs of variables mentioned above, the second period is characterised by a heterogeneous pattern over the years."

Figure 4: I find the light colours difficult to see and distinguish here, particularly for 2017 and 2018

The color table of Figure 4 has been changed to better distinguish the colors and the wording 'snowmelt' was changed to 'melt' in order to account both snow and ice melt.

[Figure]

L304: change 'the last chapter' to 'this section'

Corrected.

L309: the Copland (2011) reference talks in general terms about melting processes on glaciers, so I don't see how it supports the specific information for Baltoro Glacier provided here

Originally, we had added the reference here as Luke Copland wrote "In the spring, melting and runoff from snow and ice bodies typically occur several days to weeks after air temperatures reach 0°C as energy is initially expended on warming the snow and ice to the melting point." As this appears to be inappropriate in this specific context, we have deleted the reference as suggested.

L312: change to 'subglacial drainage system'

Corrected.

L321: the spelling of 'Schoff' should be 'Schoof'

Corrected.

L307-L329: this text is a bit difficult to follow as a single paragraph: please break up into two or more paragraphs

Thank you, you are right. For a better readability, we have divided the section into three paragraphs.

L327: do you have any evidence to prove that supraglacial lakes are drained in summer through the creation of new crevasses and cracks? E.g., high resolution imagery or field observations? Or could

other processes also be important, such as changes in surface melt, size of supraglacial channels, or formation of new supraglacial outlets from the lakes?

At Koxcar Glacier, located in the Tien Shan, we observed that a supraglacial lake drained through a crevasse. The drainage lasted only a few hours. Unfortunately, the spatial resolution of Sentinel-2 and PlanetScope is too low to observe this phenomenon on Baltoro Glacier. Though, we observed that some supraglacial lakes at Baltoro Glacier drained within 1-2 days which is an indication of crevasses as lakes with a lateral channel drain slowly. Nevertheless, supraglacial channels could not be excluded in the interpretation.

L331-L337: your descriptions in the previous sections, tables and figures don't provide clear support for the detailed sequence of events that you list here. At the moment there is lots of information presented, but it's difficult to understand how it connects to physical changes on the glacier surface. For example, you talk about detailed changes in supraglacial lake area, extent and drainage, but don't include a figure that demonstrates what these changes look like on the glacier surface. Adding a figure that shows seasonal changes in supraglacial lakes and other factors (e.g., surface melt, snow distribution) for an example region in repeat satellite imagery, for example, could therefore really help to strengthen the support for your written descriptions.

Thank you for the constructive comments. For a better linkage, we have included the references to the temporal and the spatial relationship, showing the seasonal changes of supraglacial lakes and snow melt area, in the text (Line 331). Additionally, we have referred to the figures in Wendleder et al. (2021) showing the distribution and duration of the supraglacial lakes from 2016 to 2020.

The sentences from Line 331 to 337 have been completely revised to make them easier to understand (Line 374):

"Debris covered glaciers show a different response to global warming, because ice melt is reduced by the insulating effect of the debris cover over extensive part of the ablation area (in case the debris reaches a critical thickness). Another characteristic of debris-covered glaciers is their ability to store large amounts of water in supraglacial lakes, which are suddenly released when crevasses open, often resulting in basal sliding.

The important findings of this study are 1) the period of several days with positive air temperatures in spring lead to an increase in lake rise and to a snowmelt up to 4700 m a.s.l., 2) the large melting area of snow and ice causing the spring speed-up and the high summer glacier velocities, and 3) the fact that the Baltoro glacier also shows an acceleration in fall, caused by the drained supraglacial lakes. The area affected by the melt and its effects on glacier dynamics are therefore surprisingly large. Despite the insulation due to the debris cover on the main branch of the Baltoro Glacier, the ice cliffs and supraglacial lakes act as hotspot that react sensitively to the rise in temperature and lead to an additional ice ablation."

L339-L341: you talk about snowmelt here, but typically there isn't much snow left over Baltoro Glacier in mid-July to mid-August. Instead, I expect that ice melt is the dominant mechanism, particularly over the ablation area, so you should be talking about melt originating from both ice and snow in this section.

This is correct. We have changed it to 'snow and ice melt' and corrected it throughout manuscript.

L347-L352: here and elsewhere in the Discussion you need to make a better comparison with previous work conducted on the velocity patterns of Baltoro Glacier, partly from Wendleder et al. (2018), but also from Quincey et al. (2009)

The comparison with the work of the references of Quincey et al. (2009) and Wendleder et al. (2018) has been strengthened in our study as follows:

"In the years 2017, 2019, 2020, and 2021, the velocity at Gore exceeds that at Concordia suggesting that basal sliding is greatest at this location (Fig. 2). This phenomenon was already observable in 2008-2010 and 2014 (Wendleder et al., 2018)." (Line 342)

"These high summer values were also observed in summer 2004 and 2005 (Quincey et al., 2009) and summer 2008, 2009, and 2010 (Wendleder et al., 2018)." (Line 349)

Dehecq et al. (2019) observed a small, but significant speed-up for glaciers in the Karakoram. Our observation for the Baltoro Glacier is that some years are characterized by higher glacier surface velocities, alternating with years characterized by lower values. The velocity values (or the maximum value) do not increase, but last longer in terms of time and space. Both points make a comparison with Dehecq et al. (2019) difficult and are therefore not included here. We hope that you will consider our arguments and hope for your understanding.

L354: here it sounds as if your entire study is framed about the discharge of supraglacial lakes, but this hasn't been the focus of the rest of the paper, where you also talk about snowmelt and runoff, etc. I therefore suggest rewording this.

Yes, you are right, thank you. We have reworded the sentence as follows (Line 385): "To gain a more comprehensive understanding of the processes whether and how water of snow and ice melt, supraglacial lakes, and runoff trigger or contribute to basal sliding, we conducted a spatio-temporal analysis. This involved integrating glaciological variables such as surface velocity, supraglacial lake extent, snowmelt, and runoff-index derived from Earth Observation data, along with climate variables including air temperature and precipitation obtained from the HAR dataset."

L360: change 'inefficient and efficient' to 'inefficient to efficient'

Corrected.

L360: as with earlier comment, you only talk about snowmelt here, but you can't negate ice melt

Thank you very much for spotting this inconsistency. We have changed the wording in the manuscript to "snow and ice melt".

L361-L365: I find it hard to follow the Conclusions as the statements aren't clearly supported by the information that you've presented earlier in the paper, and the sentences jump between different patterns, periods and years. For example, within these few lines you jump between fall speed up, then overall snowmelt, then spring melt, and then particular patterns in 2022. The writing needs to be clearer and more logically presented, with better reference to your supporting datasets.

Please see next specific reply.

L370: yet again, you're jumping between different topics in the same para. On L369 you seem to be referring to specific conditions in 2022, but in the next sentence you seem to be talking about multi-year velocity trends (although this is unclear, and not well supported by your data). You also need better reference to previous studies that have looked at these kinds of changes on Baltoro Glacier, such as Quincey et al. (2009), and glacier velocity changes across High Mountain Asia, such as those reported by Dehecq et al. (2018).

Thank you for your comments. Since both comments refer to the conclusion and to the same points, our answer tries to address both. For a better readability, the topics have been mentioned now chronologically, the supporting dataset has been referenced and we have added paragraphs to better structure and organize the topics. The sentence about the surface velocity has been corrected. We appreciate the idea to compare our velocity results with the results of Quincey et al. (2009) and Dehecq et al. (2018). Though, this part was shifted into the discussion in order to avoid references in the conclusion.

The conclusion was revised as followed (Line 385):

"To gain a more comprehensive understanding of whether, and how, meltwater and lake drainage events trigger or contribute to basal sliding, we conducted a spatio-temporal analysis. This involved integrating glaciological variables such as surface velocity, supraglacial lake extent, snowmelt, and runoff-index derived from Earth Observation data, along with climate variables including air temperature and precipitation obtained from the HAR dataset. The multi-parameter time series was analysed by the Pearson correlation and linear regression. Our study showed that the period of several days with positive air temperatures from the end of March to the end of April led to an abrupt transition from winter to spring and that the ablation period was influenced by continued, high air temperatures (Fig. 2). The higher air temperatures affected both snowmelt as well as supraglacial lake formation (Fig. 4a). The snowmelt of the tributary glaciers, covering up to 64 % of the total area of Baltoro Glacier (Fig. 3b) and lasting from April to November/December (Fig. 2), had the greatest impact to the basal sliding that led to the spring speed-up of 0.1-0.16 m d$^{-1}$ (30-35%) and the summer velocity increase by 0.2-0.3 m d$^{-1}$ (75-100%, Tab. A1 and A2). The subsequent transition from inefficient to efficient drainage lead to a glacier slow down. Afterwards, the supraglacial lakes with a summarized area of 3.6-5.9 km$^2$ (Fig. 3b) contribute to the fall speed-up which has a lower magnitude by 0.1-0.2 m d$^{-1}$ (20-30%) than the summer velocity peak. The discharge from snow and ice melt accounts for the largest amount of runoff (Tab. A1 and A2).

The year 2022 exhibits notable climatic impacts in the form of a sustained warm period (Tab. 1), giving rise to a series of consequential phenomena. This includes an increased number of supraglacial lakes, an advancement in the timing of maximal lake area, and a substantial amplification of snowmelt (Tab. 1). These factors collectively provoke two separate peaks in surface velocities during spring and summer (Fig. 2f), driven predominantly by the mechanism of basal sliding. Additionally, this altered dynamic triggers a more rapid transition towards an efficient drainage system (Fig. 2f)."

**References**

Dehecq, A. et al. (2019) Twenty-first century glacier slowdown driven by mass loss in High Mountain Asia. Nature Geoscience, 12, 22–27. https://doi.org/10.1038/s41561-018-0271-9

Kochtitzky, W. et al. (2022) Progress toward globally complete frontal ablation estimates of marine-terminating glaciers. Annals of Glaciology, 63(87-89), 143-142. https://doi.org/10.1017/aog.2023.35

Raymond, CF (1971) Flow in a transverse section of Athabasca Glacier, Alberta, Canada. Journal of Glaciology 10(58), 55–84. doi: 10.3189/s0022143000012995

Seroussi, H and 6 others (2011) Ice flux divergence anomalies on 79north Glacier, Greenland. Geophysical Research Letters 38(9), 1–5. doi: 10.1029/2011gl047338

Willis, I and 5 others (2003) Seasonal variations in ice deformation and basal motion across the tongue of Haut Glacier d'Arolla, Switzerland. Annals of Glaciology 36, 157–167. doi: 10.3189/172756403781816455
Citation: https://doi.org/10.5194/tc-2023-133-RC2

---

## Referee Report (RR1)

**Review of Wendleder et al.: Basal Sliding and Hydrological Drainage at Baltoro Glacier**

Wendleder et al. have addressed most of my major concerns and those of the other reviewer. I appreciate their detailed and thorough responses, and I am satisfied by their explanation of why they have not followed two of my suggestions. The Results section in particular has been reorganized and I find it much easier to follow and to identify the key findings. I have just a few remaining minor and technical corrections to suggest.

Line 6: I'm sorry, I acknowledge that I asked for more precision about the use of "runoff index", but I wonder if maybe the abstract is not the best place to add this much detail, however I do think it's important to highlight this is specifically proglacial runoff. Maybe the sentence about glacier variables could read, "For the glacier variables, we used surface velocity, supraglacial lake extent, snow and ice melt extent, and proglacial runoff index derived from Earth Observation data", and then you do not need the following sentence defining the runoff index.

Line 25: I'm sorry for being particular about the wording, but I think the wording that "sliding is initiated [...]" could be clarified further, since the authors responded to my original comment by agreeing that there is likely to be sliding in winter. Maybe it is more accurate to say that "sliding increases as the ice decouples from the bed"?

Line 31: I appreciate citing Weertman (1957) for regelation, but this explanation is still not clear to me. It is possible that you mean to explain enhanced ice-roof closure rates around obstacles (e.g., Creyts & Schoof, 2009). In that case, please provide correct and precise citations, including the listed paper and some of the references within. Otherwise, for the canonical description of channel opening and closing mechanisms, see e.g. Section 6.3 of Cuffey & Paterson (2010).

Line 32: The transition from hydrology to dynamics is quite abrupt. Given the change in topics and that this is already a long paragraph, maybe this should be split into two paragraphs to separate hydrology and dynamics.

Line 47: Reading this line, about how supraglacial lake drainage into an efficient drainage system can reduce ice velocities, has made me think: you have attributed fall speed-ups to supraglacial lake drainage. Does this explanation for fall speed-ups fit with your conceptual model of subglacial drainage development in Section 5.1? If not, could you explain this in Section 5.1 and provide some possible reasons why?

Figure 1: Please indicate in the caption what the white crosshairs represent. It would also be nice to add the flowline distance along with each place name. E.g., Urdukas / 3900 m / XX km.

Table 1: Could you explain more precisely the difference between "maximum" and "summarized" lake area in the text? My interpretation is that the "maximum" lake area refers to the maximum area that is instantaneously covered by supraglacial lakes, and that the "summarized" lake area

refs to the total area that is ever covered by lakes, but the text was not completely clear on this.

Line 308-309: The sentence "In 2018, 2020, and 2022, air temperature..." there is a comma or adjoining word missing somewhere here.

Section 5.1 and Line 359: The authors had nice and detailed responses to my earlier comments about these two sections. Could you add one to two sentences summarizing your responses to these sections? I think this would add value for the reader.

For section 5.1, maybe it would help to explicitly say that this is a hypothesis or a conceptual model. Maybe change the first sentence to something similar to, "Although each year is different, here we present a general conceptual model that explains the observed seasonal patterns".

Line 392: My apologies, I think my earlier comment was not clear. Sundal et al. (2011) show that, in a year with high surface melt rates, the peak surface velocity might be higher, but the development of efficient subglacial drainage leads to slower late-summer velocities (Figure 3). The net effect is that the overall annual ice displacement (in meters) is similar for years with low and high surface melt rates. Can you comment on whether or not you find something similar? Or does annual total displacement (or annual average velocity if you prefer to think of it that way) also increase strongly with surface melt for this glacier?

Figure 5: I like this addition to further support the discussion about changes in the time lag between peak melt and the summer velocity peak. Could the authors add just a few sentences explaining more specifically what physical processes they think could explain this dramatic change in lag time?
I am concerned that the conclusions in line 404-409 are too strong. There are only six years of data, and for processes with such a large variability, the trends are not robust. It is an interesting finding that you are correct to highlight, but I think the conclusions need to be more careful. Perhaps this could be highlighted as an interesting feature that emerged from your dense, multi-method and multi-sensor observational record that warrants further investigation.
As a technical note, the dotted lines (especially the yellow line for Urdukas) are thin and hard to see.

**References**
Creyts, T. T., and C. G. Schoof (2009), Drainage through subglacial water sheets, *J. Geophys. Res.*, 114, F04008, doi:10.1029/2008JF001215.

---

## Editor Decision (ED1)

[revised manuscript text omitted]

---

## Author Response (AR2)

**Reviewer 1:**

**Basal Sliding and Hydrological Drainage at Baltoro Glacier**

Wendleder et al. have addressed most of my major concerns and those of the other reviewer. I appreciate their detailed and thorough responses, and I am satisfied by their explanation of why they have not followed two of my suggestions. The Results section in particular has been reorganized and I find it much easier to follow and to identify the key findings. I have just a few remaining minor and technical corrections to suggest.

Dear Reviewer,

thank you very much for the helpful corrections and also for the great glaciological discussion. We really appreciate it. All the corrections and comments have been included.

All the best,

Anna Wendleder with co-authors

Line 6: I'm sorry, I acknowledge that I asked for more precision about the use of "runoff index", but I wonder if maybe the abstract is not the best place to add this much detail, however I do think it's important to highlight this is specifically proglacial runoff. Maybe the sentence about glacier variables could read, "For the glacier variables, we used surface velocity, supraglacial lake extent, snow and ice melt extent, and proglacial runoff index derived from Earth Observation data", and then you do not need the following sentence defining the runoff index.

Thank you for the suggestion. We changed it accordingly.

Line 25: I'm sorry for being particular about the wording, but I think the wording that "sliding is initiated [...]" could be clarified further, since the authors responded to my original comment by agreeing that there is likely to be sliding in winter. Maybe it is more accurate to say that "sliding increases as the ice decouples from the bed"?

You are right. We corrected it.

Line 31: I appreciate citing Weertman (1957) for regelation, but this explanation is still not clear to me. It is possible that you mean to explain enhanced ice-roof closure rates around obstacles (e.g., Creyts & Schoof, 2009). In that case, please provide correct and precise citations, including the listed paper and some of the references within. Otherwise, for the canonical description of channel opening and closing mechanisms, see e.g. Section 6.3 of Cuffey & Paterson (2010).

Thanks for the detailed explanation. We meant the canonical description and therefore referred to Cuffey and Paterson.

Line 32: The transition from hydrology to dynamics is quite abrupt. Given the change in topics and that this is already a long paragraph, maybe this should be split into two paragraphs to separate hydrology and dynamics.

Done.

Line 47: Reading this line, about how supraglacial lake drainage into an efficient drainage system can reduce ice velocities, has made me think: you have attributed fall speed-ups to supraglacial lake drainage. Does this explanation for fall speed-ups fit with your conceptual model of subglacial drainage

development in Section 5.1? If not, could you explain this in Section 5.1 and provide some possible reasons why?

You are right. In our case, the supraglacial lake drainage does not lead to the transition from inefficient to efficient subglacial drainage, but only to a speed-up in fall. We chanced the sentence in Line 47: "A few studies have observed that supraglacial lake drainage could lead to a transition from an inefficient to an efficient subglacial drainage system and hence lead to a slowdown of the glacier velocity (Vincent and Moreau, 2016; Stevens et al., 2022)". The possible reasons are discussed in section 5.2.

Figure 1: Please indicate in the caption what the white crosshairs represent. It would also be nice to add the flowline distance along with each place name. E.g., Urdukas / 3900 m / XX km.

The white crosshairs represent the coordinate grids which help to better estimate the distances. The distances along the centerline for the four locations were added in the figure.

Table 1: Could you explain more precisely the difference between "maximum" and "summarized" lake area in the text? My interpretation is that the "maximum" lake area refers to the maximum area that is instantaneously covered by supraglacial lakes, and that the "summarized" lake area refers to the total area that is ever covered by lakes, but the text was not completely clear on this.

To avoid misunderstandings, we are now using the terms "maximum area" and "total annual area" and added additionally the explanation: "The maximum area refers to the area at a specific point in time and the total summer area to the summarized area during summer."

Line 308-309: The sentence "In 2018, 2020, and 2022, air temperature..." there is a comma or adjoining word missing somewhere here.

For a better understanding, we changed the sentence to "In 2018, 2020, and 2022, air temperature and melt as well as air temperature and supraglacial lakes have a high $R^2$ (0.65-0.83), but the dependency is low (0-7 °C/km2)."

Section 5.1 and Line 359: The authors had nice and detailed responses to my earlier comments about these two sections. Could you add one to two sentences summarizing your responses to these sections? I think this would add value for the reader.

Done.

For section 5.1, maybe it would help to explicitly say that this is a hypothesis or a conceptual model. Maybe change the first sentence to something similar to, "Although each year is different, here we present a general conceptual model that explains the observed seasonal patterns".

Thank you for the hint. We changed the first sentence of this section.

Line 392: My apologies, I think my earlier comment was not clear. Sundal et al. (2011) show that, in a year with high surface melt rates, the peak surface velocity might be higher, but the development of efficient subglacial drainage leads to slower late-summer velocities (Figure 3). The net effect is that the overall annual ice displacement (in meters) is similar for years with low and high surface melt rates. Can you comment on whether or not you find something similar? Or does annual total displacement (or annual average velocity if you prefer to think of it that way) also increase strongly with surface melt for this glacier?

Thank you for the detailed explanation. We apologize that we misunderstood your question. The glacier surface velocity from 1992 to 2017 was analyzed in detail in Wendleder et al. (2018). The long time series shows that some years (2008-2010, 2015) are affected by higher averaged annual (Figure

2) and summer velocities (Figure 3), respectively. As this was a topic in the last publication, we will refrain from discussing it in more detail here.

Reference:

Wendleder, A., Friedl, P. and Mayer, C. (2018) Impacts of Climate and Supraglacial Lakes on the Surface Velocity of Baltoro Glacier from 1992 to 2017. Remote Sensing, 10 (1681), pp. 1-25. MDPI. DOI: 10.3390/rs10111681 ISSN 2072-4292.

Figure 5: I like this addition to further support the discussion about changes in the time lag between peak melt and the summer velocity peak. Could the authors add just a few sentences explaining more specifically what physical processes they think could explain this dramatic change in lag time? I am concerned that the conclusions in line 404-409 are too strong. There are only six years of data, and for processes with such a large variability, the trends are not robust. It is an interesting finding that you are correct to highlight, but I think the conclusions need to be more careful. Perhaps this could be highlighted as an interesting feature that emerged from your dense, multi-method and multi-sensor observational record that warrants further investigation. As a technical note, the dotted lines (especially the yellow line for Urdukas) are thin and hard to see.

You are fully right; the time series only covers six years and therefore no robust or significant linear trend can be derived given the high variability. However, on purpose we never use the word "trend" in the text. In the lines of the conclusion you mention (L. 406-410) we refer to this tendency as "indication" that requires a "careful interpretation". To further clarify, we added the following sentence (Line 327): "As the calculation bases only on six years, this observation requires a careful interpretation and needs further investigation." Additionally, we added an explanation for possible physical processes (Line 326): "Thus the amount of meltwater causing glacier velocity variations is reached in a shorter which will be more supported by warmer climates in the future." The lines in Figure 5 were changed and are now better recognizable. Furthermore, we have amended the figure 5 caption by emphasizing the "linear trend - to indicate the tendency only - is displayed in dotted lines.

**References**

Creyts, T. T., and C. G. Schoof (2009), Drainage through subglacial water sheets, J. Geophys. Res., 114, F04008, doi:10.1029/2008JF001215

**Reviewer 2:**

**Basal Sliding and Hydrological Drainage at Baltoro Glacier**

This paper provides an interesting and useful overview of the factors controlling velocities at Baltoro Glacier. Overall, the manuscript is much improved from the original version, and I appreciate the substantial work that the authors have made in updating it and constructively responding to the original reviewers comments. My remaining comments are all generally minor, and many reflect small technical corrections. A common issue is that 'snowmelt' is often still used in the paper, when I think that 'snow and ice melt' or 'melt' is the more accurate term to use – this is highlighted several times in my comments below.

Dear Reviewer,

thank you very much for the helpful corrections and also for finding the spelling errors. We really appreciate it. All the corrections and comments have been included.

All the best,

Anna Wendleder with co-authors

Title: the title is improved by including Pakistan in it, but it doesn't properly reflect the focus of the paper as basal sliding isn't being directly measured or quantified. Instead a title such as 'Velocity variations and hydrological drainage at Baltoro Glacier, Pakistan' would better reflect the actual content.

Thank you for your suggestion. The title was modified accordingly.

L10: here and elsewhere in the abstract you refer to 'snowmelt', but I assume that you really mean 'snow and ice melt', since there is little snow left in the summer?

Corrected in Line 10 and 12.

L18: when you refer to 'the future', do you mean in 'a future warming climate'? If so, then specify this.

Corrected.

L20: delete 'the' from 'over the glacier dynamics'

Deleted.

L22: here you refer to 'melt water', but in the next line you use 'meltwater' (no space between words). These should be consistent throughout the paper.

Thank you for the hint! We changed it to 'meltwater' and have ensured consistent spelling.

L29: change 'a efficient' to 'an efficient'

Corrected.

L32: This first para is long, so I suggest adding a para break before 'Glacier movement'

We separated it into two paragraphs.

L46: change 'inefficient drainage system' to 'inefficient subglacial drainage system' (assuming that's what you're referring to here?)

Yes, we are referring here to the subglacial drainage system. We changed it accordingly.

L58: add 'of' before 'whether'

Added.

Fig. 1 caption: add ', respectively' after 'yellow, blue, and orange'

Added.

L103: would be useful if you can include the resolution of the DEM here in m, instead of just stating 1 arc second, since the resolution of your other datasets is given in m

You are right. We have specified now the spatial resolution in meters (30 m).

L106-107: add reference to Fig. 1 here since it shows the location of these points

Thank you for the suggestion. We added the reference: "For the analysis, we extracted the surface velocity values along the glacier centerline at four different locations (Figure 1), namely at Urdukas (11 km), at the confluence with Yermanendu Glacier (16 km), Gore (23 km), and Concordia (33 km distance from terminus). The four different points were selected to best reflect the spatial variation along the glacier."

L130: change 'by 2021 and 2022' to 'to 2021 and 2022'

Changed.

L131: change to 'their aggregated area' (assuming that's what you're referring to here?)

Yes, we are referring to their aggregated area. We changed it.

L133: it would be useful to add a sentence at the start of this para to make it clear that you're referring to the proglacial stream after it exits the glacier terminus

Thank you for the suggestion. We have included the term twice to make this clearer.

L146-147: I'm unclear as to what 'assigned geographically' refers to here, and it's unclear which delta or confluence you're referring to

Apologies for the imprecise explanation. We changed the sentence to "The class "runoff" was assigned by a fixed, manually selected point on the western border of the mapping area that was always covered with water."

L173: specify that 12 pm refers to local time (e.g., rather than UTC)

We specified the time in adding the wording 'local time'.

Fig. 2a: change legend and secondary y-axis label from 'Snowmelt' to 'Snow and ice melt' (assuming that's what you mean here?).
Fig. 2: I find it difficult to distinguish between the lines plotted for Snowmelt (E), (C) and (W) in the second part of each plot since they're similar shades of grey. Can you use different colours for these lines instead?

We changed 'Snowmelt' just to 'Melt' to be consistent with Figure 4 and that the label still fits along the plot. To better distinguish the lines of the melt, we have changed the colors.

L271-272: I assume that you're referring to the transition in subglacial drainage here? If so, then specify this.

Yes, you are right. We changed it to "transition from inefficient to efficient subglacial drainage".

L275: change 'begin' to 'beginning'

Changed.

L285: I assume that you mean both snow and ice melt here, rather than just snowmelt?

Yes, we changed it to "snow and ice melt".

L290: I don't understand the units of m2. Do you mean km2?

Thanks for spotting this error! Of course, it is km2!

L293-306: for the values provided in this para specify what they refer to: e.g., write '(r=0.58-0.89)', rather than just '(0.58-0.89)'. This is because I'm a bit unclear what some of the numbers refer to. For example, on L295 you state 'high relationship (>0.6)', but is this referring to r or significance value? If it's referring to r the statement is incorrect since you say that r values ranged as low as 0.58 earlier in the sentence.

We would like to apologize for the misunderstanding. We have corrected the mistake and changed the sentence as follows: "In the two periods, the correlation of the variables temperature and melt (0.7-0.95) as well as temperature and supraglacial lake (0.58-0.89) showed a similar pattern with a correlation above 0.58 though with negligible minor changes in the Pearson correlation during both periods (0.1-0.2);"

L303-304: Pearson correlation values can vary between 0 and 1, but on these lines you say that they decreased by a value of more than 1 (e.g., <1.5). I think that you mean to say that they varied from a positive to a negative correlation (or vice versa), so this should be made clear

You are right, the sentence is misleading. We corrected it to: "Particularly noticeable was the variation between positive and negative correlation within the observation period for Urdukas in both parameters."

Fig. 3: in the legends, provide units for the velocity (presumably m d-1?). I assume that the labels should also be 'snow and ice melt', rather than snowmelt?

Thank you for spotting this! We added the unit of the glacier velocity (m/d) and changed 'snowmelt' to 'melt'.

L346: change 'snowmelt' to 'melt'

Corrected.

L370: this calculation ignores any contribution from melt of glacier ice, but in summer I expect that this is the dominant source of water on Baltoro Glacier, rather than snow melt. This would be even more true if the glacier is experiencing a negative surface mass balance. I think that these factors should at least be mentioned, and their importance briefly evaluated.

It is true that in summer the ice melt is greater than the snow melt. The crucial point is that the snowmelt starts first and thus plays the most important role in changing the dynamics. The ice melt only begins when large areas of the glacier are free of snow. At this point, however, the change in glacier velocity is already ongoing. The melt could be quantified with a degree day model. Therefore, we need the snow depth or the snow water equivalent which are not available for the region or for this study area.

L370: change 'millions' to 'million'

Corrected.

L379: I'm unclear what 'increase in lake rise' refers to. E.g., do you mean the depth of supraglacial lakes? Or perhaps the highest elevation at which they're found? Clarification is needed.

We are very sorry for the confusion. We meant the increase of the lake area. The sentence was corrected.

L387: presumably you mean 'snow and ice melt', rather than 'snowmelt'? Also elsewhere, such as L404, L406.

Corrected.

L403: make it clear here and elsewhere that you're referring to efficiency of the subglacial drainage system (i.e., as opposed to the supraglacial system)

Thank you for the suggestion. We changed it to 'subglacial drainage' throughout the manuscript.

L409: it's problematic to state that there's a higher risk of flood hazards with a faster transition to efficient subglacial drainage when you present no data to back this up. For example, it could be argued that flood hazards are actually higher when subglacial drainage is inefficient, as in this case ice-marginal lakes could build up in size if they're unable to drain all the water flowing into them – and then release catastrophically when they become overfilled. So I suggest deleting or modifying this final sentence.

We deleted the last sentence.

Tables A1 and A2: indicate in the table caption that these are r values (i.e., as opposed to r2). In the tables, I assume that you mean 'snow and ice melt' or 'melt', rather than 'snowmelt' (to match wording in Tables 2 and 3).

Thank you for spotting this. We changed both.